# The anterior cingulate cortex and its role in controlling contextual fear memory to predatory threats

**Miguel Antonio Xavier de Lima[1], Marcus Vinicius C Baldo[2], Fernando A Oliveira[3], Newton Sabino Canteras[1]\***

[1]Department of Anatomy, Institute of Biomedical Sciences, University of São Paulo, São Paulo, Brazil; [2]Department of Physiology and Biophysics, Institute of Biomedical Sciences, University of São Paulo, São Paulo, Brazil; [3]Cellular and Molecular Neurobiology Laboratory (LaNeC) - Center for Mathematics, Computing and Cognition (CMCC), Federal University of ABC, São Bernardo do Campo, Brazil

**Abstract** Predator exposure is a life-threatening experience and elicits learned fear responses to the context in which the predator was encountered. The anterior cingulate area (ACA) occupies a pivotal position in a cortical network responsive to predatory threats, and it exerts a critical role in processing fear memory. The experiments were made in mice and revealed that the ACA is involved in both the acquisition and expression of contextual fear to predatory threat. Overall, the ACA can provide predictive relationships between the context and the predator threat and influences fear memory acquisition through projections to the basolateral amygdala and perirhinal region and the expression of contextual fear through projections to the dorsolateral periaqueductal gray. Our results expand previous studies based on classical fear conditioning and open interesting perspectives for understanding how the ACA is involved in processing contextual fear memory to ethologic threatening conditions that entrain specific medial hypothalamic fear circuits.

**\*For correspondence:**
newton@icb.usp.br

**Competing interest:** The authors declare that no competing interests exist.

## Editor's evaluation

The manuscript provides a detailed circuit analysis of fear learning using a naturalistic task. The identification of anterior cingulate cortex and its input and outputs in contextual fear acquisition and expression to predator threat is an important contribution to our understanding of neural mechanism related to fear processing. The manuscript will be of interest to readers in the field of learning and memory.

## Introduction

Predator exposure is a life-threatening experience and elicits innate fear behaviors as well as learned fear responses to the context in which the predator was encountered (*Blanchard et al., 1989*; *Blanchard et al., 2001*; *Ribeiro-Barbosa et al., 2005*). Recent studies revealed a cortical network that is responsive to predatory threats and exerts a critical role in processing fear memory. Thus, the caudal prelimbic area (PL), rostral part of the anterior cingulate area (ACA), medial visual area (VISm), and the ventral part of retrosplenial area (RSPv) form a highly interconnected circuit that presents a differential increase in Fos expression in response to predator exposure. Cytotoxic lesions of the elements of this cortical circuit apparently had no impact on innate fear responses during predator exposure but had a profound impact on contextual fear memory largely disrupting learned contextual fear responses in a predator-related environment (*de Andrade Rufino et al., 2019*).

The ACA occupies a central position in this cortical network and establishes dense bidirectional connections with all other elements of the circuit. Importantly, lesions in the ACA had a larger impact on decreasing learned contextual fear responses versus lesions of the other elements of this cortical network (*de Andrade Rufino et al., 2019*). However, at this point, it is not clear how the ACA is involved in processing predator-related fear memory.

Previous studies in the literature using fear conditioning to physically aversive stimuli (i.e., foot-shock) reported important roles for the ACA in fear memory. The ACA seems to be necessary for the acquisition of contextual fear. Pretraining inactivation of the ACA blocked fear acquisition (*Tang et al., 2005*; *Bissière et al., 2008*), and pretraining activation of the ACA using a mGluR agonist (*Bissière et al., 2008*) enhanced fear learning thus suggesting an involvement of the ACA in the acquisition of fear responses.

Of relevance, the insertion of an empty temporal gap, or trace, between the CS and UCS makes learning the association critically dependent on the prefrontal cortex. Recording studies in trace fear conditioning revealed units called "bridging cells" in the prefrontal cortex that maintain firing during the trace intervals, perhaps reflecting the maintenance of attentional resources during the CS-USC interval (*Gilmartin et al., 2014*). These findings from trace fear conditioning suggest that the prefrontal cortex strengthens the predictive value of the available cue to influence memory storage. The ACA is also required for memory consolidation. Infusion of the protein-synthesis inhibitor aniso-mycin in the ACA impairs memory consolidation of recent and remote memory (*Einarsson and Nader, 2012*), and virally mediated disruption of learning-induced dendritic spine growth in the ACA impairs memory consolidation (*Vetere et al., 2011*). In fact, the ACA has been largely associated with the encoding and retrieval of contextual fear memory at remote time points (*Frankland et al., 2004*; *Kitamura et al., 2017*; *Abate et al., 2018*).

Here, we examined how that ACA mediates predator fear memory. To this end, we exposed mice to cats and investigated innate and contextual fear responses. We started by asking whether the ACA is involved in the acquisition and/or expression of predator fear memory and applied pharmacoge-netic silencing in the ACA during exposure to the predator or to the context. Next, combining retro-grade tracing and Fos protein immunostaining, we examined the pattern of activation of ACA source of inputs during acquisition and expression of predator fear responses. We could see how the ACA would be able to combine predator and contextual cues particularly during the acquisition phase. To complement these findings, during the cat exposure, we applied optogenetic inhibition to the antero-medial thalamic nucleus>ACA path, which putatively relays information related to the predator cues, and examined the effect on the acquisition of predator fear memory.

Next, using optogenetic silencing and functional tracing combining Fluoro Gold and Fos immu-nostaining, we examined how the ACA entrains selected targets to influence acquisition or expres-sion of predator fear memory. The data help to clarify how the ACA influences both the acquisition and expression of predator fear memory. Overall, the ACA offers predictive relationships between the threatening stimuli and the context to influence memory storage in amygdalar and hippocampal circuits. It also has a role in memory retrieval and the expression of contextual fear. Our results open interesting perspectives for understanding how the ACA is involved in processing contextual fear memory to predator threats as well as other ethologic threatening conditions such as those seen in the confrontation with a conspecific aggressor during social disputes.

## Results

### The ACA is involved in both the acquisition and expression of contextual fear to predator threat

To specifically evaluate the contribution of the ACA in the acquisition and expression of contextual fear to a predatory threat, we used designer receptors exclusively activated by designer drug (DREADD) to selectively silence the activity of the ACA during the cat exposure and during the exposure to the predatory context (*Figure 1*). We used adeno-associated virus (AAV) expressing Gi-coupled hM4Di fused with mCitrine fluorescent protein (AAV5-hSyn-HA-hM4D(Gi)-IRES-mCitrine). We bilaterally injected the viral vector (AAV5-hSyn-HA-hM4D(Gi)-IRES-mCitrine) or a vector expressing only the fluorescent protein for the control group (AAV5-hSyn-eGFP) into the ACA. Patch-clamp experiments showed that transfected neurons (Gi-DREADD neurons) were hyperpolarized as they underwent

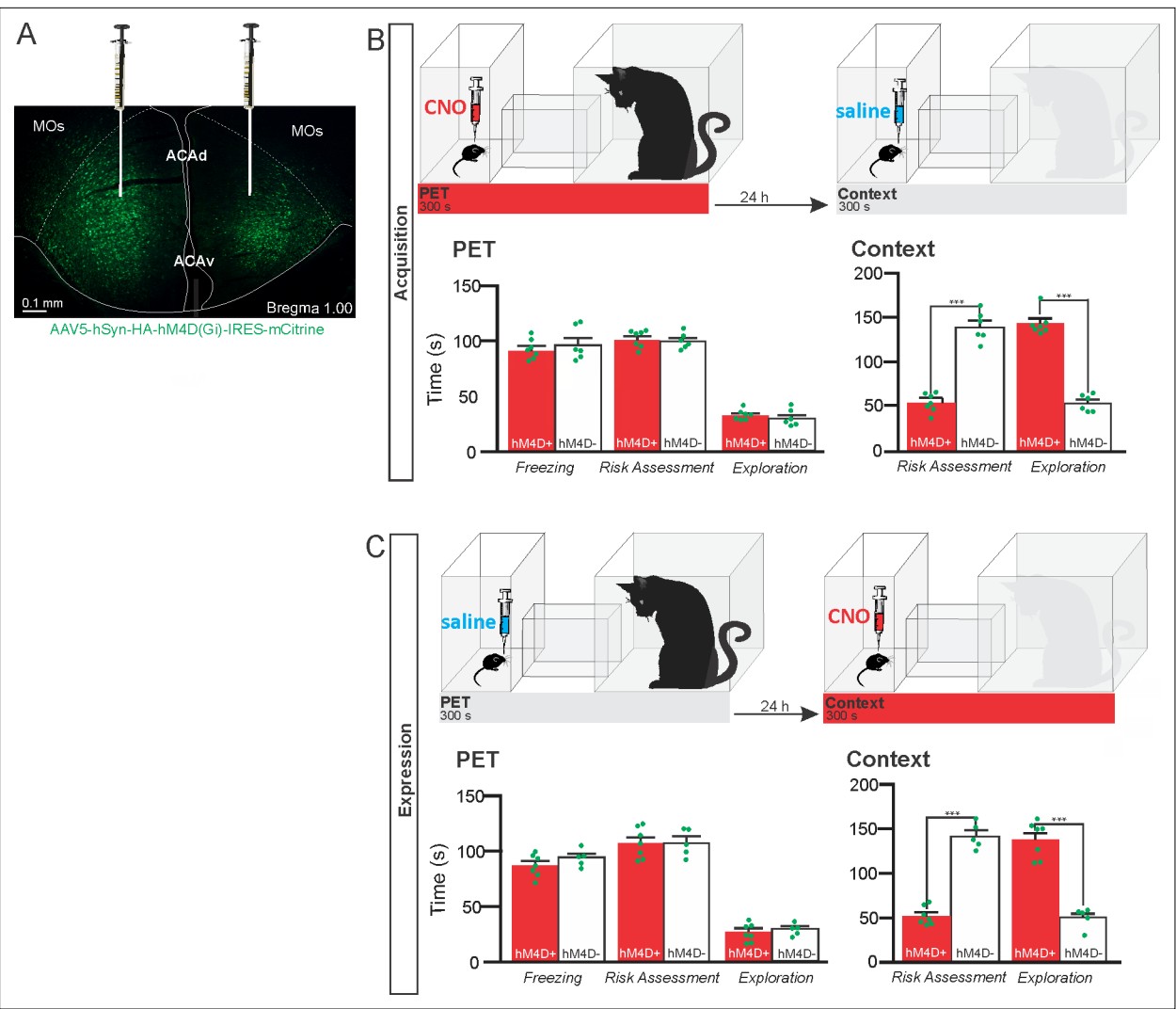

**Figure 1.** Pharmacogenetic inhibition of the ACA during acquisition and expression of contextual fear to predator threat. (**A**) Fluorescence photomicrograph illustrating the bilateral injection in the ACA of a viral vector expressing inhibitory DREADD (hM4D Gi) fused with mCitrine. (**B**) Pharmacogenetic inhibition of the ACA during the cat exposure (acquisition phase): (top) Experimental design, and (bottom) raw (dots) and mean (± SEM) values of the behavioral responses during Predator Exposure (PET) and Predatory Context (Context). For inhibition during PET- Groups: hM4D+ (n=7) and hM4D− (n=6). (**C**) Pharmacogenetic inhibition of the ACA during the predatory context (expression phase): (top) Experimental design, and (bottom) raw (dots) and mean (± SEM) values of the behavioral responses during Predator Exposure (PET) and Predatory Context (Context). For inhibition during Context − Groups: hM4D+ (n=7) and hM4D− (n=5). Data are shown as raw (dots) and mean (± SEM) values. For the freezing, a 2×2 ANOVA revealed neither main effects for the factors virus (hM4D+ and hM4D−, $F[1,21]=1.67$; $p=0.21$; $\eta^2_p=0.073$) and phase of treatment (CNO/PET and CNO/Context, $F[1,21]=1.12$; $p=0.30$; $\eta^2_p=0.051$) nor an interaction between them ($F[1,21]=0.064$; $p=0.803$; $\eta^2_p=0.003$). For the risk assessment, a three-way ANOVA revealed a significant main effect for the factor virus (hM4D+ and hM4D−, $F[1,21]=200.75$; $p<0.001$; $\eta^2_p=0.905$), no main effect for the factors phase of treatment (CNO/PET and CNO/Context, $F[1,21]=1.01$; $p=0.32$; $\eta^2_p=0.046$) and exposure (PET and Context, $F[1,21]=3.47$; $p=0.07$; $\eta^2_p=0.142$), and a significant interaction between the factors virus and exposure ($F[1,21]=144.5$; $p<0.001$; $\eta^2_p=0.873$). Post hoc pairwise comparisons (Tukey's HSD test) for the animals tested during the predatory context revealed for the hM4D+ animals that received CNO during cat exposure (**B**) and the hM4D+ group that received CNO during the predatory context (**C**) a significant decrease in risk assessment compared to the control hM4D− group (***$p<0.001$). For the relaxed exploration, a three-way ANOVA revealed a significant main effect for the factors virus (hM4D+ and hM4D−, $F[1,21]=178.74$; $p<0.001$; $\eta^2_p=0.895$) and exposure (PET and Context, $F[1,21]=429.58$; $p<0.001$; $\eta^2_p=0.953$), but no main effect for the factor phase of treatment (CNO/PET and CNO/Context, $F[1,21]=1.41$; $p=0.25$; $\eta^2_p=0.063$), and a significant interaction between the factors virus and exposure ($F[1,21]=188.22$; $p<0.001$; $\eta^2_p=0.899$). Post hoc pairwise comparisons (Tukey's HSD test) for the animals tested during the predatory context revealed for the hM4D+ animals that received CNO during cat exposure (**B**) and the hM4D+ group that received CNO during the predatory context (**C**) a significant increase in relaxed exploration compared to the control hM4D− group (***$p<0.001$). Abbreviations: ACAd, anterior cingulate area, dorsal part; ACAv, anterior cingulate area, ventral part; CNO, clozapine N-oxide; MOs, secondary motor area; PET, predator exposure test.

The online version of this article includes the following source data for figure 1:

**Source data 1.** Spreadsheet of raw values for panels (B) and (C).

extracellular CNO and presented a significant decrease in the triggering of action potentials (APs) after 10 µM CNO (*Appendix 1—figure 4*). Four weeks after the viral injection, the group of animals injected with virus expressing Gi-coupled hM4Di or the control virus received CNO intraperitoneally 30 min before the cat exposure or the exposure to the predatory context.

The results showed that compared to the control group, animals expressing Gi-coupled hM4Di treated before the cat exposure did not change innate responses but significantly reduced contextual fear responses (*Figure 1B*). Thus, during exposure to the predatory context, the animals presented a significant decrease in the risk assessment responses and an increase in fearless exploration (*Figure 1B*). Likewise, CNO treatment before the exposure to the predatory context also significantly reduced risk assessment and increased exploration in animals injected with virus expressing Gi-coupled hM4Di (*Figure 1C*). The results suggest that silencing the ACA does not influence innate fear responses but affects both the acquisition and the expression of contextual fear to predator threat.

## Pattern of activation of the different sources of inputs to the ACA during the exposure to the cat and to the predatory context

Considering the influence of the ACA in the acquisition and expression of contextual fear to predatory threat, we next examined the pattern of activation of the different sources of inputs to the ACA during the exposure to the cat and to the predatory context. To this end, animals received unilateral deposits of a retrograde tracer (Fluoro Gold) in the ACA. One week later, groups of animals were exposed either to the cat only (*Figure 2*) or to the predatory context (*Figure 3*) and perfused 90 min later. For each one of these groups, we examined the percentage of Fluoro Gold-labeled cells expressing Fos protein to see the activation pattern of the ACA inputs during the acquisition and expression of contextual fear. During the cat exposure, among the cortical inputs, the ventral retrosplenial (RSPv) and anteromedial visual (VISam) areas presented the largest percentage of FG retrogradely labeled cells expressing Fos (around 50% of the retrogradely labeled cells); the other cortical inputs to the ACA, including the medial orbital (ORBm) and prelimbic (PL) areas, displayed close to 30% of double-labeled cells (*Figure 2C*).

In addition, during cat exposure, a particularly large percentage of FG/Fos double-labeled cells were found in the central lateral (CL) and ventral anteromedial thalamic (AMv) nuclei—both of these contained more than 50% retrogradely labeled expressing Fos protein (*Figure 2C*). Notably, the CL and AMv are likely to convey information regarding the predatory threat (see Discussion). We also found that, during cat exposure, inputs to the ACA from the basolateral amygdalar nucleus, ventral hippocampus, lateral entorhinal area, and claustrum presented 25–30% FG/FOS double-labeled cells (*Figure 2C*).

Exposure to the predatory context resulted in a different profile on the activation of the inputs to the ACA (*Figure 3*). Thus, during exposure to the predatory context, all cortical inputs to the ACA as well as the inputs from the basolateral amygdala (BLA) and lateral entorhinal area presented close to 20–30% of FG-labeled cells expressing Fos protein (*Figure 3C*). In the thalamus, the AMv displayed a relatively low percentage of double-labeled cells (close to 20%), whereas the CL nucleus contained close to 40% of the retrogradely labeled cells expressing Fos protein (*Figure 3C*).

## AM>ACA projection controls the acquisition of contextual fear responses to predator threats

Considering that the AMv is one of the inputs to the ACA presenting the largest activation during cat exposure, we employed projection-based silencing approach to investigate the effect of the inhibition of the AM>ACA projection on the acquisition of contextual fear to cat exposure. For the AM>ACA projection photoinhibition, adeno-associated viral (AAV) vectors encoding halorhodopsin-3.0 fused with mCherry fluorescence protein (AAV5-hSyn-eNpHR3-mCherry) or AAV control vectors not expressing halorhodopsin-3.0 encoding mCherry fluorescence protein (AAV5-hSyn-mCherry) were injected bilaterally into the AM (*Figure 4A and B*). To test the efficiency of light-induced hyperpolarization in eNpHR3.0-expressing neurons, the patch-clamp experiments in neurons transfected with halorhodopsin revealed a robust hyperpolarization with 585 nm light on with clear linear regression due to different light intensities (*Appendix 1—figure 5*). Moreover, patch-clamp experiments tested the efficiency of pre-synaptic inhibitions and showed clear inhibition of EPSC of postsynaptic cells

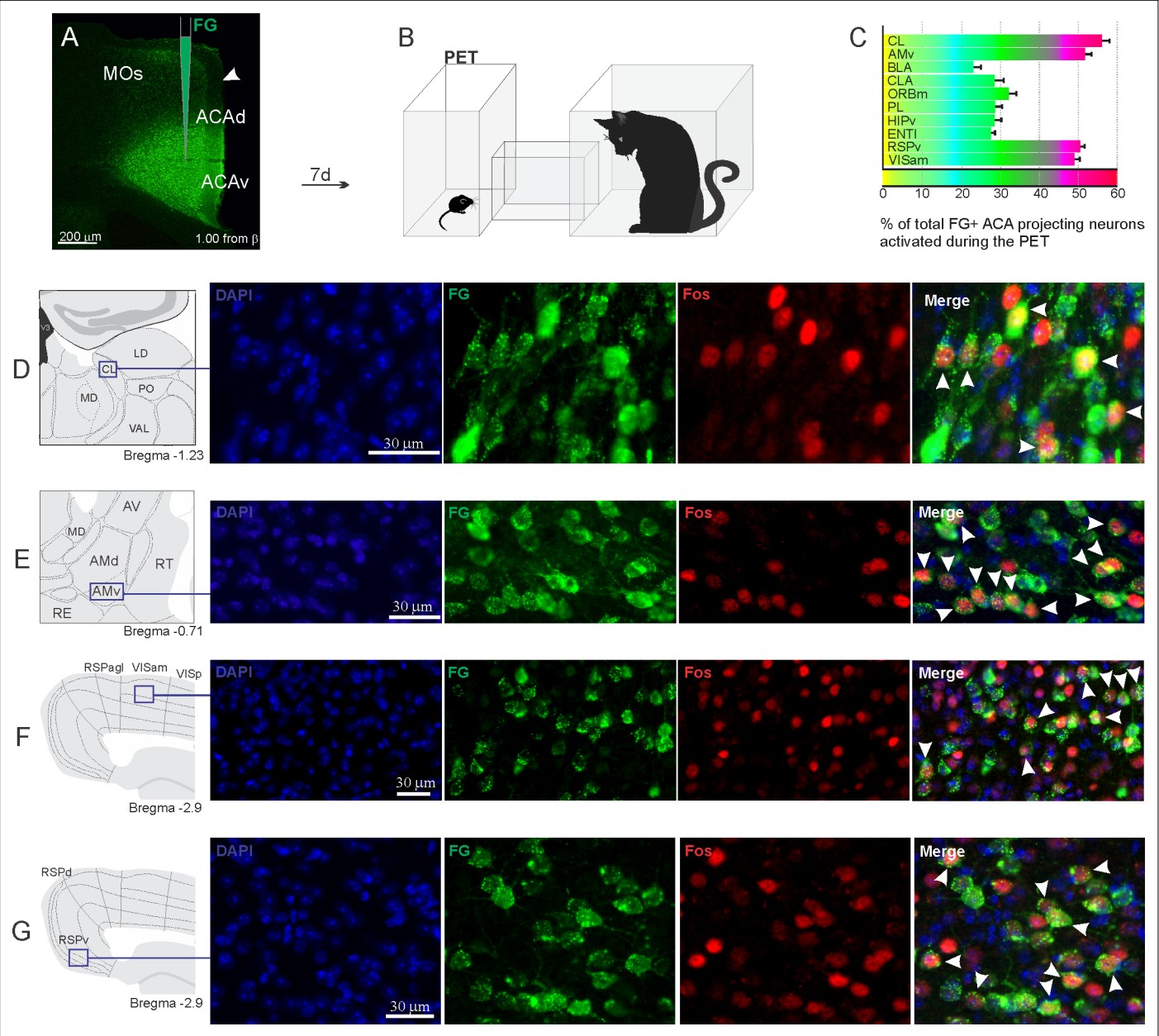

**Figure 2.** Pattern of activation of the different sources of inputs to the ACA during the exposure to the cat. Animals received unilateral deposit of a retrograde tracer (Fluoro Gold) in the ACA (n=6, **A**).After 7 days, animals were exposed to the cat (PET, **B**) and perfused 90 min after. (**C**) Bar chart presents, for each designated structure, the proportion of activated neurons (FG/Fos double-labeled cells) during the PET condition, among the total of FG retrogradely labeled cells (error bars indicate 95% confidence interval for a proportion). (**D–G**) Schematic drawings from the *Allen Mouse Brain Atlas* to show the sites containing the largest proportion of FG/Fos double-labeled cells, followed by fluorescence photomicrographs illustrating DAPI-staining, FG-labeled cells in green (Alexa 488), Fos protein positive cells labeled in red (Alexa 594) and merged view of the FG and FOS labeled cells, where arrow heads indicate FG/FOS double-labeled cells. Abbreviations: ACAd, anterior cingulate area, dorsal part; ACAv, anterior cingulate area, ventral part; AMd, anteromedial thalamic nucleus, dorsal part; AMv, anteromedial thalamic nucleus, ventral part; AV, anteroventral nucleus of thalamus; BLA, basolateral amygdalar nucleus; CL, central lateral nucleus of the thalamus; CLA, claustrum; ENTl, entorhinal area, lateral part; FG, Fluoro gold; HIPv, hippocampus, ventral part; LD, lateral dorsal nucleus of the thalamus; MD, mediodorsal nucleus of the thalamus; MOs, secondary motor area; ORBm, orbital area, medial part; PET, predator exposure test; PL, prelimbic area; PO, posterior complex of the thalamus; RE, nucleus of reuniens; RSPagl, retrosplenial area, lateral agranular part; RSPd, retrosplenial area, dorsal part; RSPv, retrosplenial area, ventral part; RT, reticular nucleus of the thalamus; VAL, ventral anterior-lateral complex of the thalamus; VISam, anteromedial visual area; VISp, primary visual area.

The online version of this article includes the following source data for figure 2:

**Source data 1.** Spreadsheet of raw values for panel (C).

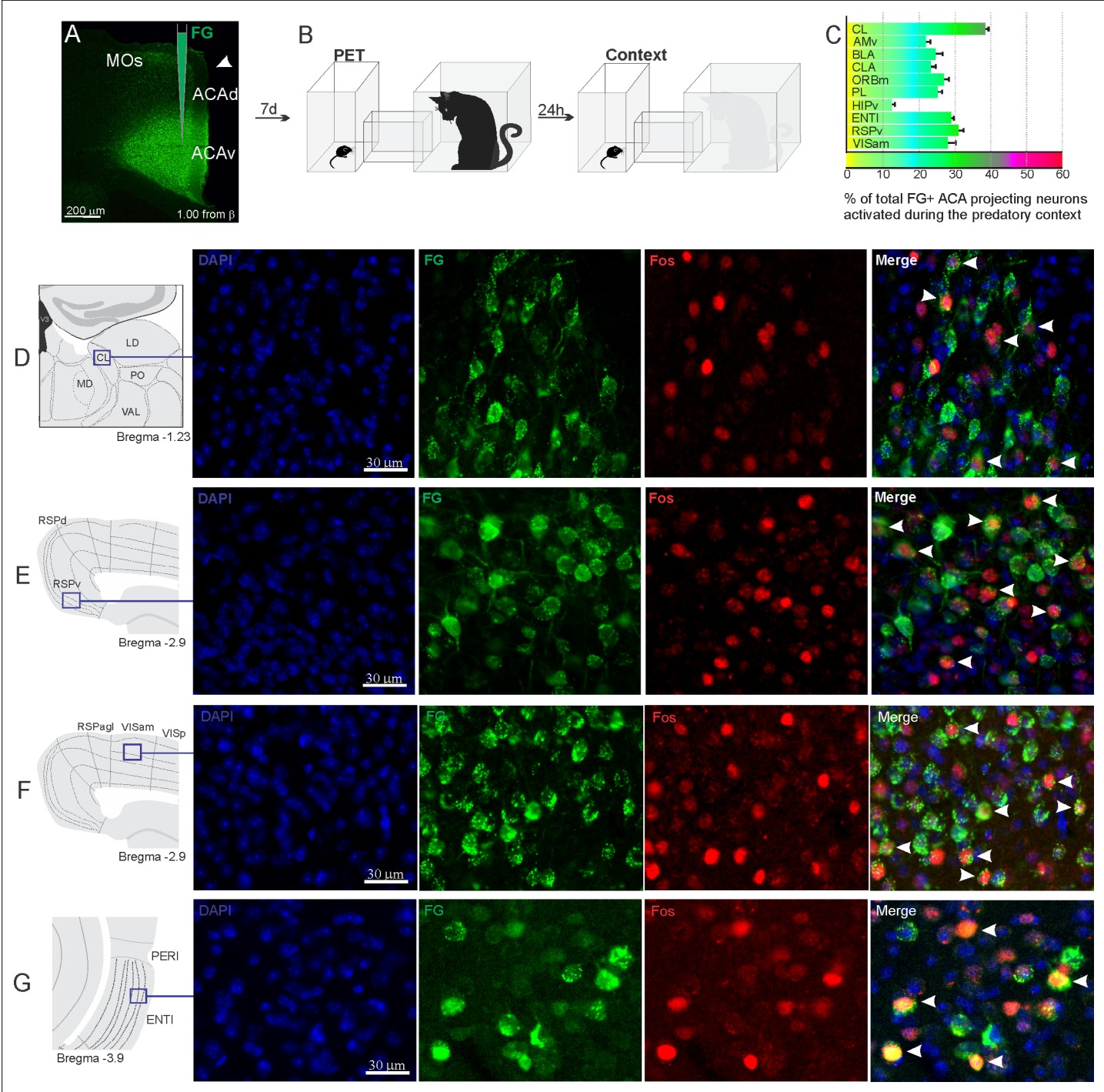

**Figure 3.** Pattern of activation of the different sources of inputs to the ACA during the exposure to predatory context. Animals received unilateral deposit of a retrograde tracer (Fluoro Gold) in the ACA (n=6, **A**).After 7 days, animals were exposed to the cat (PET) and 24 hr later to the Predatory Context (**B**) and perfused 90 min after. (**C**) Bar chart presents, for each designated structure, the proportion of activated neurons (FG/Fos double-labeled cells) during the Context condition, among the total of FG retrogradely labeled cells (error bars indicate 95% confidence interval for a proportion). (**D–G**) Schematic drawings from *Allen Mouse Brain Atlas* to show the sites containing the largest proportion of FG/Fos double-labeled cells, followed by fluorescence photomicrographs illustrating DAPI-staining, FG-labeled cells in green (Alexa 488), Fos protein positive cells labeled in red (Alexa 594), and merged view of the FG and FOS labeled cells, where arrow heads indicate FG/FOS double-labeled cells. Abbreviations: ACAd, anterior cingulate area, dorsal part; ACAv, anterior cingulate area, ventral part; AMv, anteromedial thalamic nucleus, ventral part; BLA, basolateral amygdalar nucleus; CL, central lateral nucleus of the thalamus; CLA, claustrum; ENTl, entorhinal area, lateral part; FG, Fluoro gold; HIPv, hippocampus, ventral part; LD, lateral dorsal nucleus of the thalamus; MD, mediodorsal nucleus of the thalamus; MOs, secondary motor area; ORBm, orbital area, medial part; PET, predator

*Figure 3 continued on next page*

Figure 3 continued

exposure test; PERI, perirhinal area; PO, posterior complex of the thalamus; PL, prelimbic area; RSPagl, retrosplenial área, lateral agranular part; RSPd, retrosplenial area, dorsal part; RSPv, retrosplenial area, ventral part; VAL, ventral anterior-lateral complex of the thalamus; VISam, anteromedial visual area; VISp, primary visual area.

The online version of this article includes the following source data for figure 3:

**Source data 1.** Spreadsheet of raw values for panel (C).

after illumination at 585 nm on halorhodopsin-positive fibers and no rebound excitation following halorhodopsin activation (*Appendix 1—figure 6*).

Our results showed that photoinhibition of the AM>ACA projection during cat exposure did not change innate fear responses but significantly reduced contextual fear response and impaired the acquisition of contextual fear response to a predator threat (*Figure 4E*). Thus, the animals presented a significant decrease in the risk assessment responses and an increase in fearless exploration during exposure to the predatory context (*Figure 4E*).

## Differential roles of ACA projection targets on the acquisition and expression of contextual fear to predator threat

This work used optogenetic silencing and functional tracing combining Fluoro Gold and Fos immunostaining. We examined how the ACA entrains selective targets to influence the acquisition or expression of predator fear memory. Among the ACA targets, we start by exploring projections to the BLA and the perirhinal area—these are key targets for the prefrontal cortex to influence associative plasticity and memory storage in the amygdala and hippocampus (see *Gilmartin et al., 2014*). Next, we examined the ACA projection to the postsubiculum (POST), which has a large influence on the medial performant path, and conceivably in the memory processing (*Ding, 2013*). Finally, we explored the projection to the dorsolateral periaqueductal because studies have shown its putative roles both in the acquisition and the expression of contextual fear memory to predator threat (*Cezario et al., 2008*; *de Andrade Rufino et al., 2019*); therefore, a likely ACA target to influence such responses. As illustrated in *Appendix 1—figure 7*, the pattern of ACA projection to each one of these selected targets were examined using viral tracing.

We employed projection-based silencing of the ACA projections to each one of the selected targets and tested the effect of photoinhibition during the acquisition phase and the expression of contextual fear to animals previously exposed to the cat. To this end, adeno-associated viral (AAV) vectors encoding halorhodopsin-3.0 fused with mCherry fluorescence protein (AAV5-hSyn-eNpHR3-mCherry) or AAV control vectors not expressing halorhodopsin-3.0 encoding mCherry fluorescence protein (AAV5-hSyn-mCherry) were injected bilaterally into the ACA (*Figures 5A–8A*). A 589 nm laser light was continually delivered through surgically implanted dual-fiber optic elements to each one of the ACA selected targets (*Figures 5B–8B*) during the 5 min exposure to the cat or the predatory context.

We also ran functional tracing experiments. Here, each one of the ACA selected targets received a unilateral deposit of a retrograde tracer (Fluoro Gold). After 1 week, animals were exposed either to the cat or to the predatory context and perfused 90 min after (*Figure 5E and G Figure 6E, G*; *Figure 7E, G* and *Figure 8E, G*). For each one of these targets, we examined the percentage of Fluoro Gold-labeled cells expressing Fos protein in the ACA in response to the cat exposure or exposure to the predatory context.

### ACA>BLA projection

Our results showed that photoinhibition of the ACA>BLA projection during cat exposure did not change innate fear responses but significantly reduced contextual fear responses reducing risk assessment and increasing exploration compared to the control group (*Figure 5D*). Conversely, photoinhibition of the ACA>BLA projection in eNpHR3.0-expressing mice during predatory context did not change the risk assessment and exploration times compared to the control group (*Figure 5F*). The results suggest that photoinhibition of the ACA>BLA projection impairs the acquisition but not the expression of contextual fear response to a predator threat.

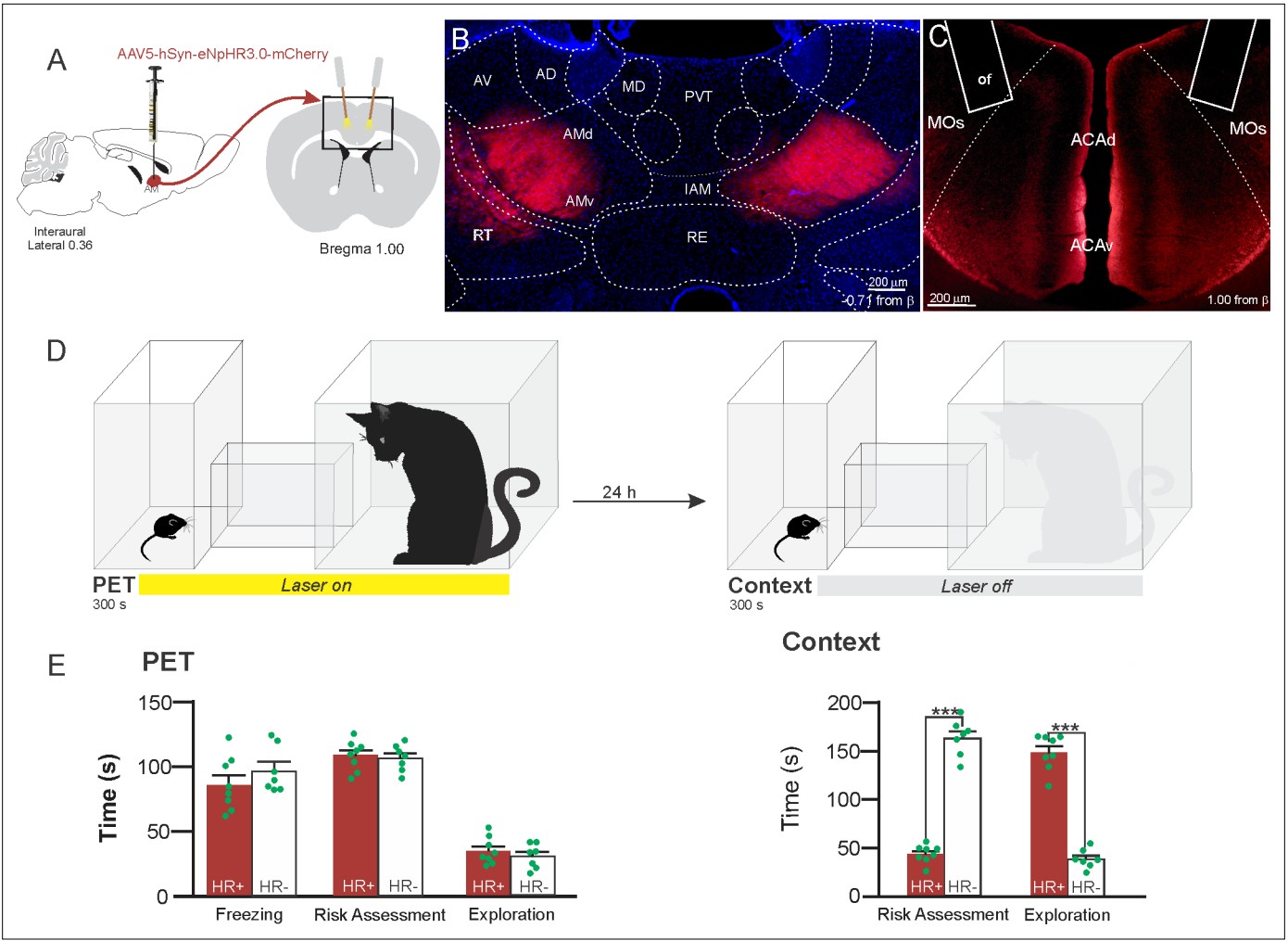

**Figure 4.** Optogenetic inhibition of anteromedial thalamic nucleus>ACA pathway during cat exposure. (**A**) Schematics showing the location of the bilateral AAV viral vector injection in the anteromedial thalamic nucleus (AM) and the position of bilateral optical fibers implanted in the ACA. (**B**) Fluorescence photomicrographs illustrating the bilateral injection in the AM of a viral vector expressing halorhodopsin-3.0 (eNpHR3.0) fused with mCherry. (**C**) Fluorescence photomicrograph illustrating the mCherry anterograde labeled projection to the ACA (*of* – the optic fibers' tips position). (**D**) Experimental design. (**E**) Raw (dots) and mean (± SEM) values of the behavioral responses during Predator Exposure (PET) and Predatory Context (Context). Data are shown as raw (dots) and mean (± SEM) values. Groups: HR+ (n=8) and HR− (n=7). For the freezing, a one-way ANOVA revealed no main effect for the factor virus (HR+ and HR−, F[1,13]=1.05; p=0.324; $\eta^2_p$=0.074). For the risk assessment, a 2×2 ANOVA revealed a main effect for the factor virus (HR+ and HR−, F[1,13]=149.79; p<0.001; $\eta^2_p$=0.920) and a significant interaction between the factors virus and exposure (F[1,13]=178.43; p<0.001; $\eta^2_p$=0.932). Post hoc pairwise comparisons (Tukey's HSD test) revealed for the HR+ animals a significant decrease in the risk assessment during the context exposure (***p<0.001) (**E**). For exploration, a 2×2 ANOVA revealed a main effect for the factor virus (HR+ and HR−, F [1,13]=102.18; p<0.001; $\eta^2_p$=0.887) and a significant interaction between the factors virus and exposure (F [1,13]=238.00; p<0.001; $\eta^2_p$=0.948). Post hoc pairwise comparisons (Tukey's HSD test) revealed for the HR+ animals a significant increase in the exploration during the context exposure (***p<0.001) (**E**). Abbreviations: ACAd, anterior cingulate area, dorsal part; ACAv, anterior cingulate area, ventral part; AD, anterodorsal nucleus of thalamus; AMd, anteromedial thalamic nucleus, dorsal part; AMv, anteromedial thalamic nucleus, ventral part; AV, anteroventral nucleus of thalamus; IAM, interanteromedial nucleus of the thalamus; MD, mediodorsal nucleus of the thalamus; MOs, secondary motor area; of, optical fiber; PET, predator exposure test; PVT, paraventricular; nucleus of the thalamus; RE, nucleus of reuniens; RT, reticular nucleus of the thalamus.

The online version of this article includes the following source data for figure 4:

**Source data 1.** Spreadsheet of raw values for panel (E).

Next, we performed a functional tracing analysis to compare the activation of the ACA>BLA pathway during the cat exposure and the predatory context. Animals that received FG injection in the BLA underwent statistical analysis (Chi-square test) of the functional tracing. The results revealed a significant difference between the proportion of Fos/FG double-labeled cells in the cat exposure condition as compared to the predatory context condition ($\chi^2$[0.05,1]=84.12; p<0.001) (**Figure 5E**

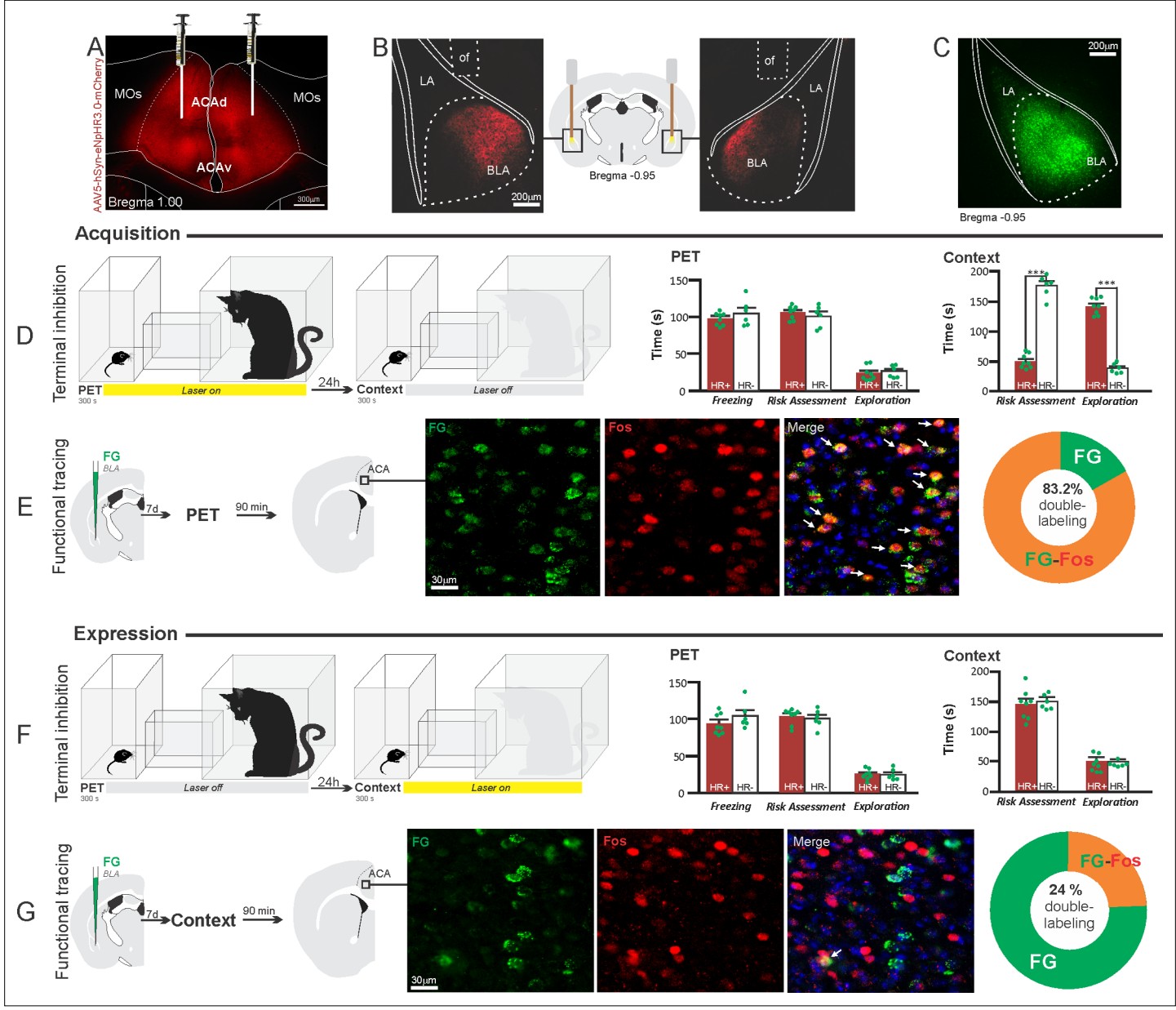

**Figure 5.** Optogenetic silencing and functional tracing of the ACA>BLA pathway during the acquisition and expression of contextual fear to predator threat. (**A**) Fluorescence photomicrograph illustrating the bilateral injection in the ACA of a viral vector expressing halorhodopsin-3.0 (eNpHR3.0) fused with mCherry. (**B**) Schematic drawing (center) and fluorescence photomicrographs showing the ACA projections to the BLA and location of bilateral optical fibers implanted close to the basolateral amygdala (*of* – optic fibers' tips position). (**C**) Fluorescence photomicrograph illustrating the FG injection in the BLA for the functional tracing. (**D, F**) Optogenetic silencing of the ACA>BLA pathway during the cat exposure (**D**) or predatory context (**F**). Experimental design (on the left) and raw (dots) and mean (± SEM) values of the behavioral responses during Predator Exposure (PET) and Predatory Context (on the right). For silencing during PET condition – Groups: HR+ (n=8) and HR− (n=6); for silencing during Context condition – Groups: HR+ (n=8) and HR− (n=6). Data are shown as raw (dots) and mean (± SEM) values. For the freezing, a 2×2 ANOVA revealed neither main effects for the factors virus (HR+ and HR−, $F[1,24]=2.98$; p=0.097; $\eta^2_p=0.110$) and phase of treatment (Photoinhibition/PET and Photoinhibition/Context, $F[1,24]=0.20$; p=0.656; $\eta^2_p=0.008$) nor an interaction between them ($F[1,24]=0.053$; p=0.819; $\eta^2_p=0.002$). For the risk assessment, a three-way ANOVA revealed significant main effects for the factors virus (HR+ and HR−, $F[1,24]=78.6$; p<0.001; $\eta^2_p=0.766$), phase of treatment (Photoinhibition/ PET and Photoinhibition/Context, $F[1,24]=23.83$; p<0.001; $\eta^2_p=0.498$) and exposure (PET and Context, $F[1,24]=35.38$; p<0.001; $\eta^2_p=0.596$), as well as a significant three-way interaction among these factors ($F[1,24]=45.74$; p<0.001; $\eta^2_p=0.656$). For the animals tested during the predatory context, post hoc pairwise comparisons (Tukey's HSD test) revealed for the HR+ animals that received photoinhibition during cat exposure a significant decrease in risk assessment compared to the control HR− group (***p<0.001) (**D**), whereas the HR+ group that received photoinhibition during the context did not differ from the control HR− group (p=0.99) (**F**). For the exploration, a three-way ANOVA revealed significant main effects for the factors virus (HR+ and HR−, $F[1,24]=106.76$; p<0.001; $\eta^2_p=0.816$), phase of treatment (Photoinhibition/PET and Photoinhibition/Context, $F[1,24]=70.91$; p<0.001;

*Figure 5 continued on next page*

*Figure 5 continued*

$\eta^2_p$=0.747) and exposure (PET and Context, F[1,24]=256.10; p<0.001; $\eta^2_p$=0.914), as well as a significant three-way interaction among these factors (F[1,24]=93.76; p<0.001; $\eta^2_p$=0.796). For the animals tested during the predatory context, post hoc pairwise comparisons (Tukey's HSD test) revealed for the HR+ animals that received photoinhibition during cat exposure a significant increase in the relaxed exploration compared to the control HR− group (***p<0.001) (**D**), whereas the HR+ group that received photoinhibition during the context did not differ from the control HR− group (p=0.99) (**F**). (**E, G**) Functional tracing of the ACA>BLA pathway during the cat exposure (acquisition phase, **E**) and the predatory context (expression phase, **G**). Left – Experimental design: unilateral FG injection in the BLA, and 7 days later perfusion 90 min after the cat exposure (n=4, **E**) or the context exposure (n=4, **G**). Center – Fluorescence photomicrographs illustrating, in the ACA, FG-labeled cells in green (Alexa 488), Fos protein positive cells labeled in red (Alexa 594) and merged view of the FG and FOS labeled cells (arrows indicate FG/FOS double-labeled cells). Right – Graphic representation of the percentage of FG/Fos double-labeled cells in the ACA. Abbreviations: ACAd, anterior cingulate area, dorsal part; ACAv, anterior cingulate area, ventral part; BLA, basolateral amygdalar nucleus; FG, Fluoro gold; LA, lateral amygdalar nucleus; MOs, secondary motor area; of, optical fiber; PET, predator exposure test.

The online version of this article includes the following source data for figure 5:

**Source data 1.** Behavior data – spreadsheet of raw values for panels (D) and (F).

*and G*). The ACA contained a significantly larger percentage of double-labeled Fos/FG cells in animals exposed to the cat (83.2%, double-to-chance ratio=1.94, 95%confidence interval [CI] [1.42, 2.47]) compared to those exposed to the predatory context (24%, double-to-chance ratio=0.81, 95% CI [0.61, 1.02]) (*Figure 5E and G*). This suggests a larger engagement of the ACA cells projecting to the BLA during the acquisition compared to the expression of contextual fear anti-predatory responses.

## ACA>PERI projection

The results showed that photoinhibition of the ACA>PERI projection during cat exposure did not change innate fear responses but significantly reduced contextual fear responses reducing risk assessment and increasing exploration compared to the control group (*Figure 6D*). Conversely, photoinhibition of the ACA>PERI projection in eNpHR3.0-expressing mice during predatory context did not change the risk assessment and exploration times versus the control group (*Figure 6F*). The results suggest that silencing the ACA>PERI projection impairs the acquisition but not the expression of contextual fear response to a predator threat.

Next, we performed a functional tracing analysis to compare the activation of the ACA>PERI pathway during the cat exposure and the predatory context. For the animals that received the retrograde FG injection in the PERI, the statistical analysis (Chi-square test) of the functional tracing results revealed a significant difference between the proportion of Fos/FG double-labeled cells in the cat exposure condition versus the predatory context condition ($\chi^2$[0.05,1]=56.552; p<0.001) (*Figure 6E and G*); here, the ACA contained a significantly larger percentage of double-labeled Fos/FG cells in animals exposed to the cat (48.5%, double-to-chance ratio=1.10, 95% CI [0.84, 1.37]) compared to those exposed to the predatory context (15.2%, double-to-chance ratio=0.64 95% CI [0.53, 0.75]) (*Figure 6E and G*). This suggests a larger engagement of the ACA cells projecting to the PERI during the acquisition compared to the expression of contextual fear responses.

## ACA>POST projection

Compared to the control group, NpHR3.0-expressing mice that received photoinhibition of the ACA>POST projection during cat exposure or predatory context did not change the risk assessment responses and fearless exploration during exposure to the environment previously visited by a predator (*Figure 7D and F*). The results suggest that silencing the ACA>POST projection apparently had no effect on the acquisition and expression of contextual fear to a predator threat. Note, in these experiments, the position of the optic fibers was consistent across the experiments and suitable to silence the ACA>POST pathway (see *Appendix 1—figure 10*).

Next, we performed a functional tracing analysis to compare the activation of the ACA>POST pathway during the cat exposure and the predatory context. Functional tracing in POST FG-injected animals revealed that the ACA contained a relatively low percentage of double-labeled Fos/FG cells in response either to the cat (21.4%, double-to-chance ratio = 0.52 95% CI [0.38, 0.65]) or the predatory context (19.1%, double-to-chance ratio = 0.78 95% CI [0.68, 0.88]) (*Figure 7E and G*). The statistical analysis (Chi-square test) revealed no significant difference between the proportion of Fos/FG double-labeled cells in the cat exposure condition versus the predatory context condition

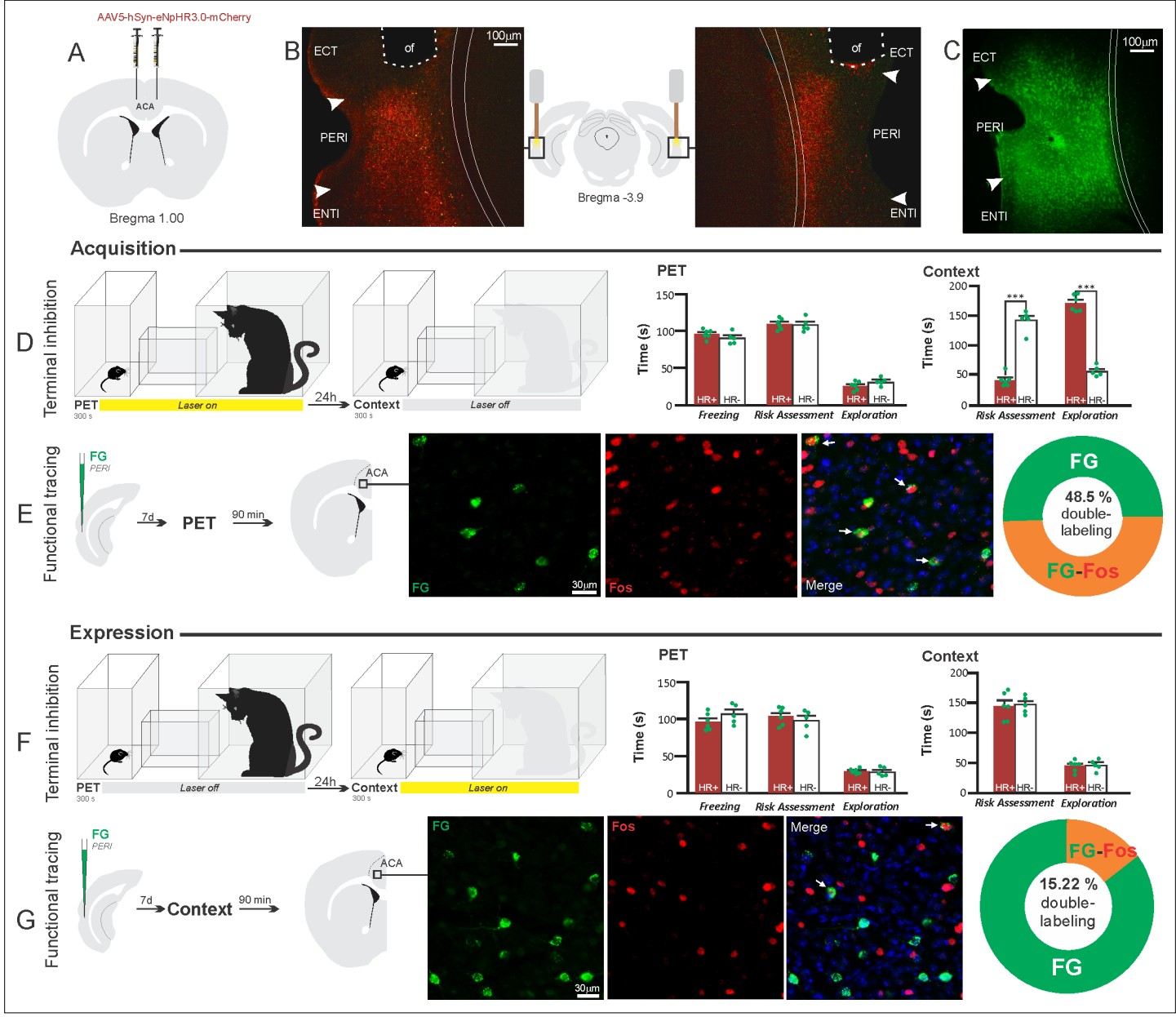

**Figure 6.** Optogenetic silencing and functional tracing of the ACA>PERI pathway during the acquisition and expression of contextual fear to predator threat. (**A**) Schematic drawing illustrating the bilateral injection in the ACA of a viral vector expressing halorhodopsin-3.0 (eNpHR3.0) fused with mCherry. (**B**) Schematic drawing (center) and fluorescence photomicrographs showing the ACA projections to the PERI and location of bilateral optical fibers implanted close to the PERI (*of* – optic fibers' tips position). (**C**) Fluorescence photomicrograph illustrating the FG injection in the PERI for the functional tracing. (**D, F**) Optogenetic silencing of the ACA>PERI pathway during the cat exposure (**D**) or predatory context (**F**). Experimental design (on the left), and raw (dots) and mean (± SEM) values of the behavioral responses during Predator Exposure (PET) and Predatory Context (on the right). For silencing during PET condition – Groups: HR+ (n=6) and HR− (n=5); for silencing during Context condition – Groups: HR+ (n=6) and HR− (n=5). Data are shown as raw (dots) and mean (± SEM) values. For the freezing, a 2×2 ANOVA revealed neither main effect for the factor virus (HR+ and HR−, F[1,18]=0.259; p=0.617; $\eta^2_p$=0.014) and the factor phase of treatment (Photoinhibition/PET and Photoinhibition/Context, F[1,18]=4.46; p=0.049; $\eta^2_p$=0.198), and no significant interaction between them (F[1,18]=3.24; p=0.088; $\eta^2_p$=0.152). For the risk assessment, a three-way ANOVA revealed significant main effects for the factors virus (HR+ and HR−, F[1,18]=61.56; p<0.001; $\eta^2_p$=0.773) and phase of treatment (Photoinhibition/PET and Photoinhibition/Context, F[1,18]=63.25; p<0.001; $\eta^2_p$=0.778), but not for the factor exposure (PET and Context, F[1,18]=7.69; p=0.0125; $\eta^2_p$=0.299). There was a significant three-way interaction among these factors (F[1,18]=19.36; p<0.001; $\eta^2_p$=0.518). For the animals tested during the predatory context, post hoc pairwise comparisons (Tukey's HSD test) revealed for the HR+ animals that received photoinhibition during cat exposure a significant decrease in risk assessment compared to the control HR− group (***p<0.001) (**D**), whereas the HR+ group that received photoinhibition during the context did not differ from the control HR− group (p=0.96) (**F**). For the exploration, a three-way ANOVA revealed significant main effects for the factors virus (HR+ and HR−,

*Figure 6 continued on next page*

*Figure 6 continued*

F[1,18]=135.44; p<0.001; $\eta^2_p$=0.883), phase of treatment (Photoinhibition/PET and Photoinhibition/Context, F[1,18]=204.85; p<0.001; $\eta^2_p$=0.919) and exposure (PET and Context, F[1,18]=343.98; p<0.001; $\eta^2_p$=0.950), as well as a significant three-way interaction among these factors (F[1,18]=127.74; p<0.001; $\eta^2_p$=0.876). For the animals tested during the predatory context, post hoc pairwise comparisons (Tukey's HSD test) revealed for the HR+ animals that received photoinhibition during cat exposure a significant increase in the relaxed exploration compared to the control HR− group (***p<0.001) (**D**), whereas the HR+ group that received photoinhibition during the context did not differ from the control HR− group (p=0.99) (**F**). (**E, G**) Functional tracing of the ACA>PERI pathway during the cat exposure (acquisition phase, **E**) and the predatory context (expression phase, **G**). Left – Experimental design: unilateral FG injection in the PERI, and 7 days later perfusion 90 min after the cat exposure (n=4, **E**) or the context exposure (n=4, **G**). Center – Fluorescence photomicrographs illustrating, in the ACA, FG-labeled cells in green (Alexa 488), Fos protein positive cells labeled in red (Alexa 594), and merged view of the FG and FOS labeled cells (arrows indicate FG/FOS double-labeled cells). Right – Graphic representation of the percentage of FG/Fos double-labeled cells in the ACA. Abbreviations: ACA, anterior cingulate area; ECT, ectorhinal area; FG, Fluoro gold; of, optical fiber; PET, predator exposure test; ENTl, entorhinal area, lateral part; PERI, perirhinal area.

The online version of this article includes the following source data for figure 6:

**Source data 1.** Behavior data – spreadsheet of raw values for panels (D) and (F).

($\chi^2$[0.05,1]=0.05206; p>0.75) (*Figure 7E and G*). Thus, there is a relatively small engagement of the ACA cells projecting to the POST during both the acquisition and expression of contextual fear anti-predatory responses.

## ACA>PAGdl projection

These results revealed that photoinhibition of the ACA>PAGdl projection during cat exposure did not change innate or contextual fear responses (*Figure 8D*). Conversely, compared to the control group, photoinhibition of the ACA>PAGdl projection during exposure to the predatory context in eNpHR3.0-expressing mice yielded a significant decrease in the risk assessment and an increase in exploration (*Figure 8F*). The results showed that photoinhibition of the ACA>PAGdl projection had no effect on the acquisition but impaired the expression of contextual fear to a predator threat.

Next, we performed a functional tracing analysis to compare the activation of the ACA>PAGdl pathway during the cat exposure and the predatory context. For the animals that received the retrograde FG injection in the PAGdl, the statistical analysis (Chi-square test) of the functional tracing results revealed a significant difference between the proportion of Fos/FG double-labeled cells in the cat exposure condition as compared to the predatory context condition ($\chi^2$[0.05, 1]=79.624; p<0.001) (*Figure 8E and G*). The ACA contained a significantly larger percentage of double-labeled Fos/FG cells in animals exposed to the predatory context (75.2%, double-to-chance ratio=2.66 95% CI [2.30, 3.02]) compared to those exposed to the cat (29.9%, double-to-chance ratio = 0.67 95% CI [0.43, 0.90]) (*Figure 8E and G*). This suggests a larger engagement of the ACA cells projecting to the PAGdl during the expression compared to the acquisition of contextual fear responses.

## Discussion

Combining the use of pharmaco- and optogenetic circuit manipulation tools and functional tracing, we untangled the ACA role in processing contextual fear memory to a predator threat.

To test the acquisition and expression of predator contextual fear responses, we used a two-phase paradigm composed of cat exposure (acquisition phase), and, on the following day, exposure to the context where the predator had been previously encountered (expression phase) (see Appendix – Behavioral Protocol). Notably, the defensive responses in mice differ somewhat from those seen in rats exposed to cats and then to the predator-associated context. During cat exposure, rats spend close to 90% of the time freezing and 2% of the time risk assessing the environment *Ribeiro-Barbosa et al., 2005*; mice froze for approximately 30% of the time and risk assessed the environment for close to 30% of the time (see Appendix – Behavioral Protocol). During exposure to the predatory context, rats only froze a little and then spent time risk assessing the predator-related context for 70% of the time (*Ribeiro-Barbosa et al., 2005*), which is different from mice that presented no freezing and a relatively weaker risk assessment response—close to 50% of the time (see *Appendix 1—table 1*).

To uncover whether the ACA response is specific to threat and threat paired context, we examined the ACA activity in four different conditions: exposure to the cat, the cat-related context, the novel non-threat stimulus (plush cat), and the context paired with non-threat stimulus (see Appendix

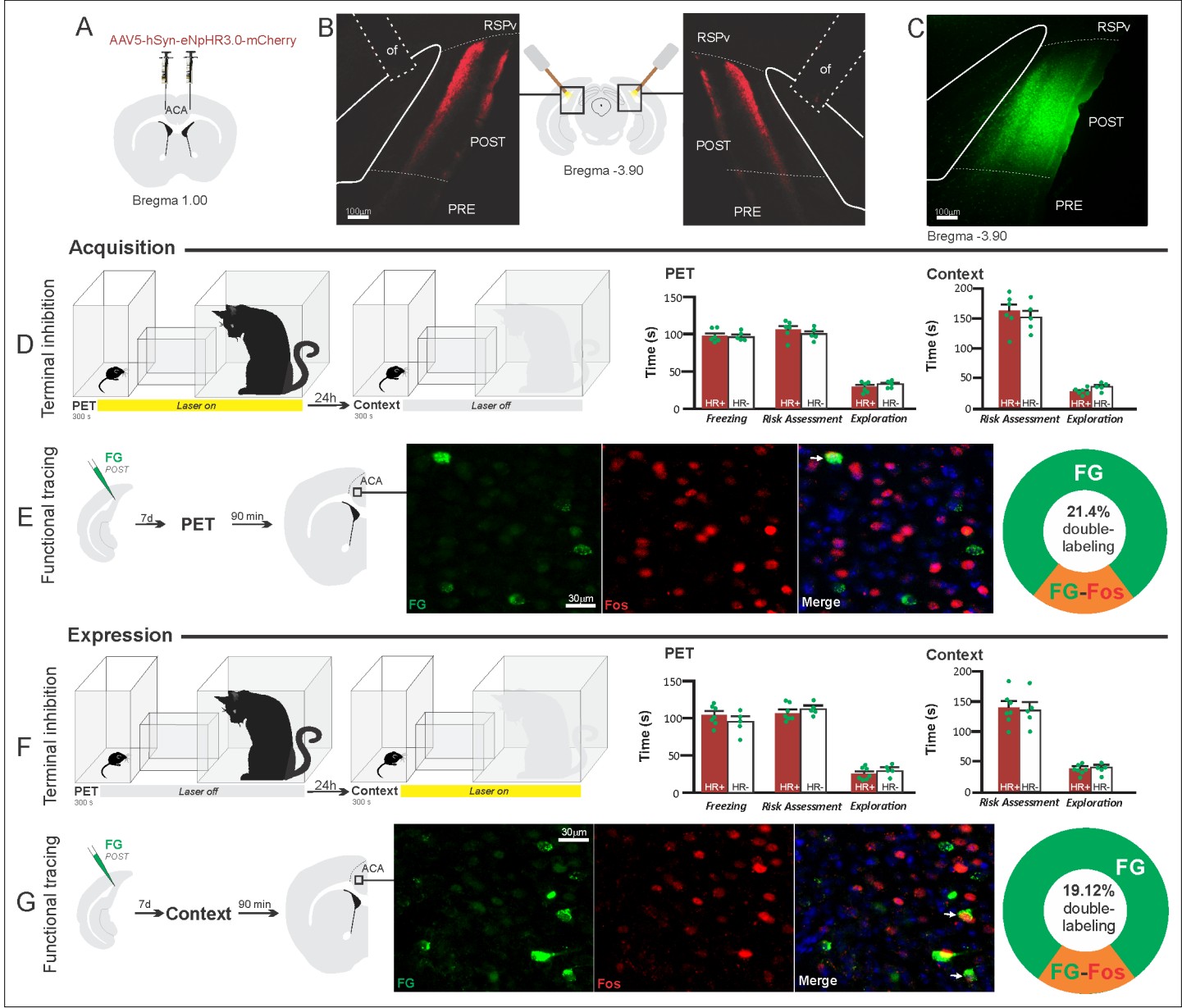

**Figure 7.** Optogenetic silencing and functional tracing of the ACA>POST pathway during the acquisition and expression of contextual fear to predator threat. (**A**) Schematic drawing illustrating the bilateral injection in the ACA of a viral vector expressing halorhodopsin-3.0 (eNpHR3.0) fused with mCherry. (**B**) Schematic drawing (center) and fluorescence photomicrographs showing the ACA projections to the POST and location of bilateral optical fibers implanted close to the POST (*of* – optic fibers' tips position). (**C**) Fluorescence photomicrograph illustrating the FG injection in the POST for the functional tracing. (**D, F**) Optogenetic silencing of the ACA>POST pathway during the cat exposure (**D**) or predatory context (**F**). Experimental design (on the left) and raw (dots) mean (± SEM) values of the behavioral responses during Predator Exposure (PET) and Predatory Context (on the right). For silencing during PET condition – Groups: HR+ (n=6) and HR− (n=5); for silencing during Context condition – Groups: HR+ (n=7) and HR− (n=5). Data are shown as raw (dots) and mean (± SEM) values. For the freezing, a 2×2 ANOVA revealed neither main effects for the factors virus (HR+ and HR−, F[1,19]=1.08; p=0.311; $\eta^2_p$=0.054) and phase of treatment (Photoinhibition/PET and Photoinhibition/Context, F[1,19]<0.001; p=0.997; $\eta^2_p$<0.001) nor an interaction between them (F[1,19]=0.85; p=0.367; $\eta^2_p$=0.043). For the risk assessment, a three-way ANOVA revealed no main effects for the factors virus (HR+ and HR−, F[1,19]=0.37; p=0.551; $\eta^2_p$=0.019) and phase of treatment (Photoinhibition/PET and Photoinhibition/Context, F[1,19]=1.64; p=0.215; $\eta^2_p$=0.079), and a significant main effect for the factor exposure (PET and Context, F[1,19]=39.56; p<0.001; $\eta^2_p$=0.675). There was no significant three-way interaction among these factors (F[1,19]=0.018; p=0.894; $\eta^2_p$<0.001). Post hoc pairwise comparisons (Tukey's HSD test) for the animals tested during the predatory context revealed no difference for the HR+ and HR− animals that received photoinhibition during cat exposure (p=0.994) or during the context (p=1) (**D, F**). For the exploration, a three-way ANOVA revealed no significant main effects for the factors virus (HR+ and HR−, F[1,19]=2.63; p=0.121; $\eta^2_p$=0.122) and phase of treatment (Photoinhibition/PET and Photoinhibition/Context, F[1,19]=0.19; p=0.663; $\eta^2_p$=0.10), and a significant main effect for the factor exposure (PET and Context, F[1,19]=18.18; p<0.001; $\eta^2_p$=0.489). There was no significant three-way interaction among these factors

*Figure 7 continued*

(F[1,19]=1.21; p=0.285; $\eta^2_p$=0.059). Post hoc pairwise comparisons (Tukey's HSD test) for the animals tested during the predatory context revealed no difference for the HR+ and HR− animals that received photoinhibition during cat exposure (p=0.728) or during the context (p=0.999) (**D, F**). (**E, G**) Functional tracing of the ACA>POST pathway during the cat exposure (acquisition phase, **E**) and the predatory context (expression phase, **G**). Left – Experimental design: unilateral FG injection in the POST, and 7 days later perfusion 90 min after the cat exposure (n=4, **E**) or the context exposure (n=4, **G**). Center – Fluorescence photomicrographs illustrating, in the ACA, FG-labeled cells in green (Alexa 488), Fos protein positive cells labeled in red (Alexa 594), and merged view of the FG and FOS labeled cells (arrows indicate FG/FOS double-labeled cells). Right – Graphic representation of the percentage of FG/Fos double-labeled cells in the ACA. Abbreviations: ACA, anterior cingulate area; FG, Fluoro gold; of, optical fiber; PET, predator exposure test; POST, postsubiculum; PRE, presubiculum; RSPv, retrosplenial area, ventral part.

The online version of this article includes the following source data for figure 7:

**Source data 1.** Behavior data – spreadsheet of raw values for panels (D) and (F).

– Comparison of ACA activity). This analysis revealed that ACA Fos expression in response to the cat and cat-related context was significantly higher relative to its expression level in response to exposure to a novel non-threat stimulus and to the context paired with a non-threat stimulus (see *Appendix 1— figure 3*).

Pharmacogenetic inhibition of the ACA during cat exposure did not change innate responses but significantly reduced contextual fear responses. This suggests that the ACA is involved in the acquisition of predator-related contextual fear responses. In line with these results, studies using fear conditioning to physically aversive stimuli (i.e., foot shock) have also supported the idea that the ACA appears to be necessary for the acquisition of contextual fear (*Tang et al., 2005*; *Bissière et al., 2008*).

Our functional tracing analysis combining retrograde tracer in the ACA and Fos immunostaining revealed that the ACA is particularly influenced by afferent sources of inputs conveying contextual information and predator cues during the acquisition phase. Among the cortical inputs, the ventral retrosplenial (RSPv) and medial visual (VISm) areas presented the largest percentage of FG retrogradely labeled cells expressing Fos (around 50% of the retrogradely labeled cells). The retrosplenial cortex is critical for the recognition of an environment directly paired with an aversive event (*Robinson et al., 2018*). Recent studies using two-photon imaging revealed that landmark cues served as dominant reference points and anchored in the retrosplenial cortex the spatial code, which is the result of local integration of visual, motor, and spatial information (*Fischer et al., 2020*). The VISm is necessary for the visuospatial discrimination used in the integration of allocentric visuospatial cues (*Sánchez et al., 1997*; *Espinoza et al., 1999*) and may as well be involved in integrating contextual cues. Therefore, the results suggest that cortical fields projecting to the ACA particularly active during the acquisition phase of predator fear memory are involved in computing contextual landmarks.

In the thalamus, our functional tracing revealed that the CL and AMv nuclei contained more than 50% of the FG retrogradely labeled cells expressing Fos during the acquisition phase of predator fear memory. Both the CL and AMv are likely to convey information regarding the predatory threat. The CL receives projections from the dorsal part of the periaqueductal gray (*Kincheski et al., 2012*), and the AMv receives substantial inputs from the dorsal premammillary nucleus (*Canteras and Swanson, 1992*). The dorsal premammillary nucleus is part of the predator-responsive circuit of the medial hypothalamus (*Gross and Canteras, 2012*) and is the most responsive site to a live predator or its odor (*Cezario et al., 2008*; *Dielenberg et al., 2001*).

In order to test whether the information relayed through the projection from the anteromedial thalamic nucleus (AM) to the ACA influences the acquisition of predator fear memory, we silenced the AM>ACA pathway during cat exposure. Photoinhibition of the AM>ACA projection during cat exposure did not change innate fear responses but significantly reduced contextual fear responses. This suggests that silencing the AM>ACA projection impairs the acquisition of contextual fear response to a predator threat. In line with this result, studies from our group have shown that pharmacological inactivation of the AM drastically reduced the acquisition but not the expression of contextual fear responses to predator threats (*de Lima et al., 2017*). Therefore, the AM>ACA pathway does not seem involved in the expression of contextual fear. It is noteworthy that anterior thalamic lesions slow down the acquisition of contextual fear conditioning but do not affect contextual fear memory tested in the short term (*Dupire et al., 2013*; *Marchand et al., 2014*).

Our findings collectively support the idea that the ACA integrates contextual and predator cues during the acquisition phase to provide predictive relationships between the context and the

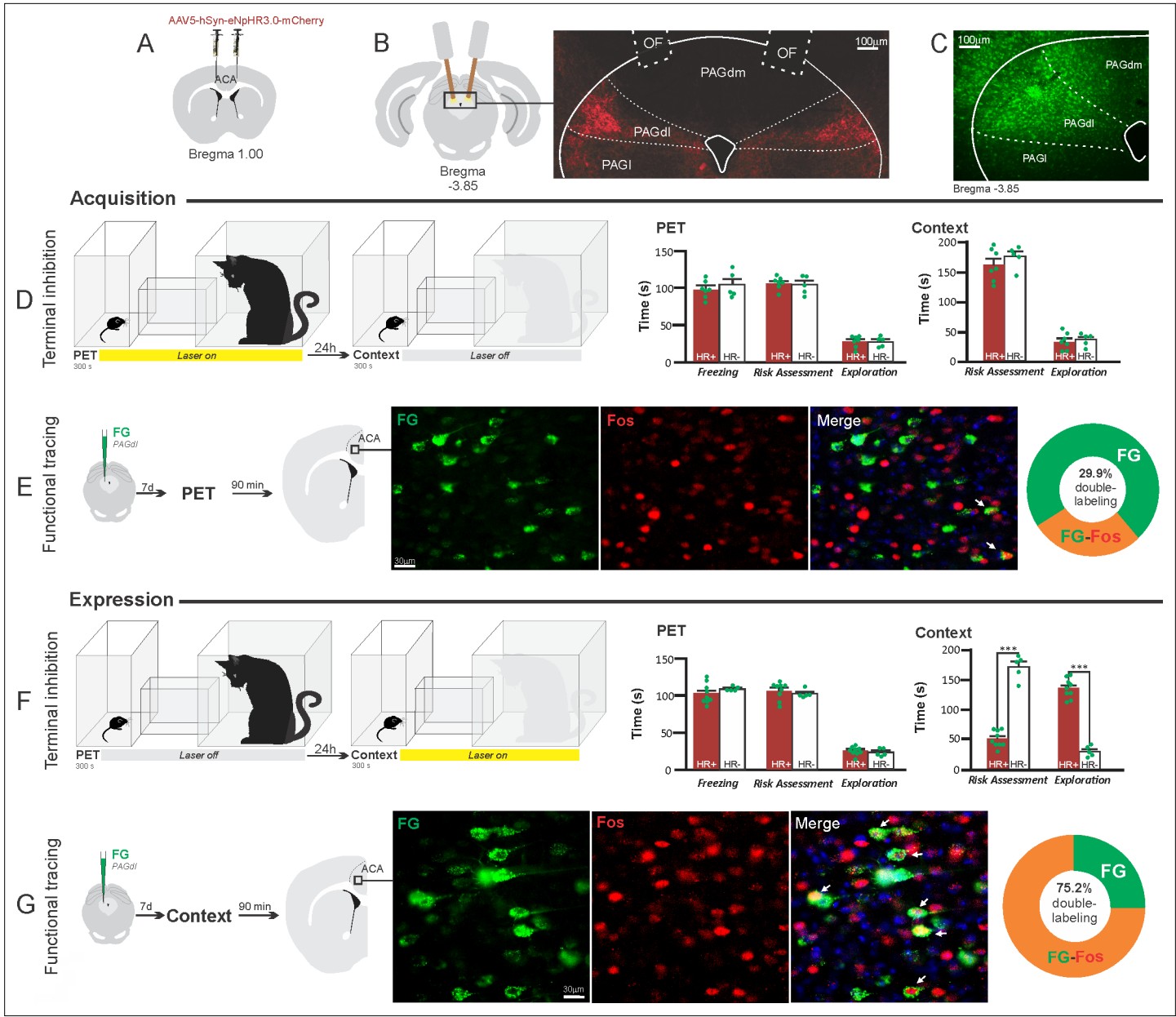

**Figure 8.** Optogenetic silencing and functional tracing of the ACA>PAGdl pathway during the acquisition and expression of contextual fear to predator threat. (**A**) Schematic drawing illustrating the bilateral injection in the ACA of a viral vector expressing halorhodopsin-3.0 (eNpHR3.0) fused with mCherry. (**B**) Schematic drawing (left) and fluorescence photomicrograph (right) showing the ACA projections to the PAGdl and location of bilateral optical fibers implanted close to the PAGdl (*of* – optic fibers' tips position). (**C**) Fluorescence photomicrograph illustrating the FG injection in the PAGdl for the functional tracing. (**D, F**) Optogenetic silencing of the ACA>PAGdl pathway during the cat exposure (**D**) or the predatory context (**F**). Experimental design (on the left) and raw (dots) and mean (± SEM) values of the behavioral responses during Predator Exposure (PET) and Predatory Context (on the right). For silencing during PET condition – Groups: HR+ (n=7) and HR− (n=5); for silencing during Context condition – Groups: HR+ (n=9) and HR− (n=5). Data are shown as raw (dots) and mean (± SEM) values. For the freezing, a 2×2 ANOVA revealed neither main effects for the factors virus (HR+ and HR−, F[1,22]=1.40; p=0.249; $\eta^2_p$=0.059) and phase of treatment (Photoinhibition/PET and Photoinhibition/Context, F[1,22]=1.11; p=0.302; $\eta^2_p$=0.048) nor an interaction between them (F[1,22]=0.01; p=0.923; $\eta^2_p$<0.001). For the risk assessment, a three-way ANOVA revealed significant main effects for the factors virus (HR+ and HR−, F[1,22]=38.05; p<0.001; $\eta^2_p$=0.634), phase of treatment (Photoinhibition/PET and Photoinhibition/Context, F[1,22]=33.25; p<0.001; $\eta^2_p$=0.602) and exposure (PET and Context, F[1,22]=77.22; p<0.001; $\eta^2_p$=0.778), as well as a significant three-way interaction among these factors (F[1,22]=45.23; p<0.001; $\eta^2_p$=0.673). For the animals tested during the predatory context, post hoc pairwise comparisons (Tukey's HSD test) revealed for the HR+ animals that received photoinhibition during the context a significant decrease in risk assessment compared to the control HR− group (***p<0.001) (**F**), whereas the HR+ group that received photoinhibition during cat exposure did not differ from the control HR− group (p=0.852) (**D**). For the exploration, a three-way ANOVA revealed significant main effects for the factors virus (HR+ and HR−, F[1,22]=90.20;

*Figure 8 continued on next page*

*Figure 8 continued*

p<0.001; $\eta^2_p$=0.804), phase of treatment (Photoinhibition/PET and Photoinhibition/Context, F[1,22]=64.93; p<0.001; $\eta^2_p$=0.747) and exposure (PET and Context, F[1,22]=103.45; p<0.001; $\eta^2_p$=0.825), as well as a significant three-way interaction among these factors (F[1,22]=62.25; p<0.001; $\eta^2_p$=0.739). For the animals tested during the predatory context, post hoc pairwise comparisons (Tukey's HSD test) revealed for the HR+ animals that received photoinhibition during the context a significant increase in the exploration compared to the control HR− group (***p<0.001) (**F**), whereas the HR+ group that received photoinhibition during the cat exposure did not differ from the control HR− group (p=0.99) (**D**). (**E, G**) Functional tracing of the ACA>PAGdl pathway during the cat exposure (acquisition phase, **E**) and the predatory context (expression phase, **G**). Left – Experimental design: unilateral FG injection in the PAGdl, and 7 days later perfusion 90 min after the cat exposure (n=4, **E**) or the context exposure (n=4, **G**). Center – Fluorescence photomicrographs illustrating, in the ACA, FG-labeled cells in green (Alexa 488), Fos protein positive cells labeled in red (Alexa 594), and merged view of the FG and FOS labeled cells (arrows indicate FG/FOS double-labeled cells). Right – Graphic representation of the percentage of FG/Fos double-labeled cells in the ACA. Abbreviations: ACA, anterior cingulate area; FG, Fluoro gold; of, optical fiber; PAGdl, periaqueductal gray, dorsolateral part; PAGdm, periaqueductal gray, dorsomedial part; PAGl, periaqueductal gray, lateral part; PET, predator exposure test.

The online version of this article includes the following source data for figure 8:

**Source data 1.** Behavior data – spreadsheet of raw values for panels (D) and (F).

---

threatening stimuli and influence memory storage. The ventral hippocampus and BLA appear as critical sites for memory storage of contextual fear to predator threats. Ventral hippocampal cytotoxic lesions significantly reduced conditioned defensive behaviors during re-exposure to the predator-associated context (*Pentkowski et al., 2006*). Likewise, cytotoxic NMDA lesions in the BLA impaired contextual fear responses to predator threat (*Martinez et al., 2011*; *Bindi et al., 2018*). In line with these ideas, recent findings from our lab indicate that cycloheximide (a protein synthesis inhibitor) injection in the BLA or the ventral hippocampus impairs contextual fear responses to predatory threat (F. Reis and N.S. Canteras personal observations). Therefore, we investigated how the ACA would facilitate associative plasticity and memory storage in the BLA and hippocampus. To this end, we used optogenetic silencing and functional tracing combining Fluoro Gold and Fos immunostaining to examine how the ACA entrains selective targets to influence acquisition or expression of predator fear memory.

Among the ACA targets, we start by exploring projections to the BLA and the perirhinal region (PERI). Previous model of the prefrontal regulation of memory formation suggests that the prefrontal cortex is likely to influence memory storage in the amygdala and hippocampus using two branches (*Gilmartin et al., 2014*). One direct branch is to the BLA and conveys information about the predictive value of the relevant clues during learning. The other branch is to the perirhinal cortices, which occupies a strategic position in this network influencing both the hippocampus and the amygdala. Tract-tracing studies revealed that the PERI provides dense projections to the BLA and the hippocampal formation (*Shi and Cassell, 1999*). Photoinhibition of both ACA>BLA and ACA>PERI pathways during cat exposure significantly reduced contextual fear responses thus suggesting a role in the acquisition of predator fear memory. In contrast, photoinhibition of both pathways did not influence the expression of contextual fear memory. Corroborating these findings, our functional tracing revealed that both ACA>BLA and ACA>PERI paths are significantly more active during the acquisition compared to the expression of predator contextual fear memory. Triple retrograde tracing experiments revealed that the ACA cells projecting to the BLA and the PERI were mostly found in the supragranular layers (layers II and III), and close to 50% of these cells provided a branched projection to the BLA and PERI (see in the Appendix, Triple retrograde tracing and *Appendix 1—figure 9*).

The POST in turn is necessary for normal acquisition of contextual and auditory fear conditioning (*Robinson and Bucci, 2012*). Thus, we tested the ACA>POST pathway as a putative candidate to influence predator fear memory. However, the ACA>POST pathway's photoinhibition did not impair either the acquisition or the expression of contextual fear memory. Notably, we showed that the position of the optic fibers was consistent across these experiments and suitable to silence the ACA>POST pathway. In favor of the lack of effect of the ACA>POST pathway, our functional tracing examined the percentage of Fos positive cells in the ACA projecting to the POST and revealed a relatively low percentage of double-labeled cells both in the acquisition and the expression of predatory contextual fear memory. Moreover, among the ACA paths that received optogenetic silencing in the present investigation, we noted that the ACA>POST projection presented the smallest density of terminals (see in the Appendix, Quantification of ACA terminal fields and *Figure 8*).

Pharmacogenetic inhibition of the ACA during the exposure to the predatory context significantly reduced contextual fear responses. This result further suggests that the ACA is also involved in the expression of predator-related contextual fear responses. Exposure to the predatory context yielded a different activation pattern among the sources of inputs to the ACA compared to those seen during cat exposure. Unfortunately, our functional tracing does not offer any plausible hint regarding the putative source of inputs to the ACA driving retrieval of memory traces to influence contextual fear expression. In this regard, we note that the ACA is involved in the consolidation of contextual fear conditioning to physically aversive stimuli (*Einarsson and Nader, 2012*). Future work could investigate whether the ACA is also involved in the consolidation of contextual fear memory to predator threat and how this consolidation would influence the retrieval predator fear memory in the ACA.

The dorsolateral periaqueductal gray (PAGdl) appears to be a likely ACA target to influence the expression of contextual fear responses to predator threats. Exposure to the predatory context yields a significant Fos expression in the dorsal PAG (*Cezario et al., 2008*)—a region where electrical, pharmacological, and optogenetic stimulation have been shown to produce freezing, flight, and risk assessment behavior in the absence of a predatory threat (*Bittencourt et al., 2004*; *Assareh et al., 2016*; *Deng et al., 2016*). The present findings showed that photoinhibition of the ACA>PAGdl pathway during cat exposure did not change innate or contextual fear responses. Conversely, photoinhibition of the ACA>PAGdl projection during predatory context altered contextual fear responses. In line with these findings, the functional tracing revealed that the ACA contained a significantly larger proportion of double-labeled Fos/FG cells for the PAGdl FG injected animals in response to the context versus the cat exposure. Collectively, our findings revealed that the ACA>PAGdl pathway is significantly more active during exposure to the predatory context and influences the expression of contextual fear to predator threat. Triple retrograde tracing experiments revealed that ACA cells projecting to the PAGdl are in the infragranular layers (layers V and VI) and do not overlap with the cells projecting to the BLA or PERI (see in the Appendix, Triple retrograde tracing and *Appendix 1—figure 9*).

Finally, it is noteworthy that all ACA paths involved in the acquisition or expression of contextual fear responses were differentially activated between acquisition and expression of fear memory and are unlikely to provide a condition to produce a state-dependent effect on contextual fear memory. However, regardless of endogenous activity patterns, there is a hypothetical possibility of a disruptive effect if memory is encoded in the absence of a given pathway and tested in its presence. In this case, it is important to bear in mind that as much as the optogenetic silencing of a given pathway is effective, it is unlikely to provide a complete inactivation of the entire pathway.

Overall, the ACA can provide predictive relationships between the context and the predator threat and influences fear memory acquisition through projections to the BLA and PERI and the expression of contextual fear through projections to the PAGdl. These findings may be applied in more general terms to understand memory processing of fear threats entraining hypothalamic circuits (i.e., such as the predator- and conspecific-responsive circuits underlying predatory and social threats, respectively) that engage the AMv>ACA pathway (Gross and Canteras, 2019). Notably, both the predator- and conspecific-responsive hypothalamic circuits comprise the dorsal premammillary nucleus that projects densely to the AMv nucleus (*Canteras and Swanson, 1992*), and therefore, engage the AMv>ACA pathway. Of relevance, previous studies showed that the anteromedial thalamus' cytotoxic lesions impair contextual fear in a social defeat-associated context (*Rangel et al., 2018*). Thus, the present results open interesting perspectives for understanding how the ACA is involved in processing contextual fear memory to predator threats as well as other ethologic threatening conditions such as those seen in confrontation with a conspecific aggressor during social disputes.

We observed that male and female mice present similar defensive responses during exposure to the predator or predatory context (*Appendix 1—table 1*). Here, it is important to note that regardless of similar innate and contextual fear responses seen in males and females, the underlying circuits are not necessarily the same. In fact, in the literate, there is a lack of investigation on the neural circuits mediating anti-predatory responses in female rodents, and this is certainly a limitation of the present investigation. A recent study carried out in male and female mice examining the role of the dorsal premammillary nucleus—a key element of the hypothalamic predator-responsive circuit—on coordination of anti-predatory responses did not report differences between sexes (*Wang et al., 2021*). However, sex differences are expected in terms of the responsivity of elements of defensive circuits, especially considering that critical sites in these circuits, such as the hypothalamus, hippocampus, and

amygdala, are responsive to gonadal steroid hormones (*Simerly et al., 1990*). At this point, further studies are needed to investigate sex differences in the circuits mediating contextual fear memory to predatory threats.

## Materials and methods

**Key resources table**

| Reagent type (species) or resource | Designation | Source or reference | Identifiers | Additional information |
|---|---|---|---|---|
| Strain, strain background (*Mus musculus*) | C57BL/6 | Local breeding facilities | | |
| Strain, strain background (*Felis catus*) | Male cat | Domestic | | |
| Transfected construct (adeno-associated virus) | AAV5-hSyn-HA-hM4D(Gi)-IRES-mCitrine | Addgene plasmid #50464 (RRID: Addegene_50464) | | Dr. Bryan Roth; titer≥7×10$^{12}$ vg/ml |
| Transfected construct (adeno-associated virus) | AAV5- hSyn-eGFP | Addgene viral prep #50465 | | Dr. Bryan Roth; titer≥7×10$^{12}$ vg/ml |
| Transfected construct (adeno-associated virus) | AAV5-hSyn-eNpHR3-mCherry | University of North Carolina, Vector Core | | Titer ≥1×10$^{13}$ vg/ml |
| Transfected construct (adeno-associated virus) | AAV5-hSyn-mCherry | University of North Carolina, Vector Core | | Titer ≥7×10$^{12}$ vg/ml |
| Transfected construct (adeno-associated virus) | AAV-Retro-Ef1a-mcherry- IRES-Cre | Addgene catalog #55632-AAVrg | | Karl Deisseroth; Titer ≥7×10$^{12}$ vg/ml |
| Antibody | (Polyclonal rabbit anti-c-Fos) | PC-38; Calbiochem-Millipore (RRID:AB_2106755) | | (1:20,000) |
| Antibody | (Polyclonal anti-rabbit Alexa 594 goat IgG) (H+L) | Invitrogen (RRID:AB_2534079) | | (1:500) |
| Antibody | (Polyclonal rabbit anti-FG antibody) | Chemicon International, CA | | (1:5000) |
| Antibody | (Polyclonal anti-rabbit Alexa 488 Goat IgG) (H+L) | Invitrogen (RRID:AB_143165) | | (1:1000) |
| Antibody | (Polyclonal anti-rabbit Alexa 405 goat IgG) (H+L) | Invitrogen (RRID:AB_221605) | | (1:500) |
| Antibody | (Polyclonal goat anti-CTb) | List Biological Laboratories, Campbell, CA | | (1:20,000) |
| Antibody | (Polyclonal anti-goat Alexa 488 donkey IgG) (H+L) | Invitrogen (RRID:AB_2534102) | | (1:500) |
| Chemical compound, drug | Fluoro Gold | Fluorochrome Inc, CO | FG | |
| Chemical compound, drug | Isoforine | Cristália Laboratories, SP, Brazil | | |
| Chemical compound, drug | Clozapine-N-oxide | Tocris Bioscience, UK | CNO | |
| Chemical compound, drug | Subunit B of the cholera toxin | List Biological Laboratories, Campbell, CA | CTb | |
| Software, algorithm | Fiji | https://imagej.net/software/fiji/ (RRID:SCR_002285) | | |

*Continued on next page*

*Continued*

| Reagent type (species) or resource | Designation | Source or reference | Identifiers | Additional information |
|---|---|---|---|---|
| Software, algorithm | BORIS software | DOI: 10.1111/2041–210X.12584 (RRID:SCR_021434) | | Behavior Observation Research Interactive Software |
| Software, algorithm | pClamp 10.7 | Molecular Devices, USA (RRID:SCR_011323) | | |
| Software, algorithm | Clampfit 10.7 | Molecular Devices, USA | | |
| Software, algorithm | CorelDraw 2018 | https://www.coreldraw.com/br/pages/coreldraw-2018/ (RRID:SCR_014235) | | |
| Software, algorithm | Statistica 7.0 | StatSoft (RRID:SCR_014213) | | |
| Other | DAPI | Sigma-Aldrich | | 1:20,000 |

## Animals

Adult male (n=192) and female (n=6) mice, C57BL/6 weighing approximately 28 g, were used in the present study. The mice were obtained from local breeding facilities and were kept individually housed under controlled temperature (23°C) and illumination (12 hr cycle) in the animal quarters with free access to water and a standard laboratory diet. All experiments and conditions of animal housing were carried out under institutional guidelines (Colégio Brasileiro de Experimentação Animal [COBEA]) and were in accordance with the NIH Guide for the Care and Use of Laboratory Animals (NIH Publications No. 80-23, 1996). All of the experimental procedures had been previously approved by the Committee on the Care and Use of Laboratory Animals of the Institute of Biomedical Sciences, University of São Paulo, Brazil (Protocol No. 085/2012). Experiments were always planned to minimize the number of animals used and their suffering.

## Sterotaxic surgery, viral injections, and optical fiber implantation

Mice were anesthetized in a box saturated with Isoforine (Cristália Laboratories, SP, Brazil) and then immediately positioned on a stereotaxic instrument (Kopf Instruments, CA). The anesthesia was maintained with 1–2% Isoforine/oxygen mix and body temperature was controlled with a heating pad. Viral vectors were injected with a 5-µl Hamilton Syringe (Neuros Model 7000.5 KH). Injections were delivered at a rate of 5 nl/min using a motorized pump (Harvard Apparatus). The needle was left in place for 5 min after each injection to minimize upward flow of viral solution after raising the needle. For the photoinhibition experiments, immediately after the viral injections, optical fibers (Mono Fiber-optic Cannulae 200/230-0.48, Doric Lenses Inc, Quebec, Canada) were implanted and fixed onto animal skulls with dental resin and micro screws (DuraLay, IL). The animals were allowed 1 week to recover from the surgery, and 3 weeks later, we started the behavioral experiments.

For the pharmacogenetic inhibition, ACA was injected bilaterally with 150 nl of AAV5-hSyn-HA-hM4D(Gi)-IRES-mCitrine (Dr. Bryan Roth; Addgene plasmid #50464) or AAV5-hSyn-eGFP (titer≥7×10$^{12}$ vg/ml; Addgene viral prep #50465-AAV5) as control. For the photoinhibition of the AM>ACA projection, mice were bilaterally injected into the AM (AP –0.7, ML ± 0.5, DV –3.7) with 20 nl of AAV5-hSyn-eNpHR3-mCherry (titer≥1×10$^{13}$ vg/ml; University of North Carolina, Vector Core) or AAV5-hSyn-mCherry (control virus; titer≥7×10$^{12}$ vg/ml; University of North Carolina, Vector Core), and optical fibers were bilaterally implanted into the ACA (at a 10° angle from the vertical axis; AP +1.0, ML ± 0.7, DV –0.6). For the photoinhibition of the ACA projections to selected targets during acquisition and expression of contextual fear responses, mice were bilaterally injected into the ACA (AP +1.0, ML ± 0.3, DV –1.1) with 80 nl of AAV5-hSyn-eNpHR3-mCherry (titer≥1 ×10$^{13}$ vg/ml; University of North Carolina, Vector Core) or AAV5-hSyn-mCherry (control virus; titer≥7×10$^{12}$ vg/ml; University of North Carolina, Vector Core), and optical fibers were bilaterally implanted into the BLA (AP –0.95, ML ± 3.3, DV –3.3), PERI (AP –3.9, ML ± 0.5 from the lateral skull surface, DV –0.5 from the local brain surface), POST (at a 20° angle from the vertical axis; AP –3.9, ML ± 2.7, DV –0.75), or PAGdl (at a 15° angle from the vertical axis; AP –3.85, ML ±0.9, DV –1.8). For tracing the ACA projections, mice were unilaterally injected into the ACA (AP +1.0, ML ±0.3, DV –1.1) with 80 nl of AAV5-hSyn-eNpHR3-mCherry

(titer≥1×10$^{13}$ vg/ml; University of North Carolina, Vector Core). For the experiments with triple retrograde labeling, mice were unilaterally injected into the dorsolateral periaqueductal gray (PAGdl; AP –3.85, ML +0.45, DV –2.1) with 50 nl of AAV-Retro-Ef1a-mcherry- IRES-Cre (titer≥7×10$^{12}$ vg/ml; Karl Deisseroth; Addgene catalog #55632-AAVrg).

## Experimental apparatus and behavioral procedures

All the testing procedures were carried out during the light phase of the cycle. The experimental apparatus was made of clear Plexigas and consisted of a Box 1 (with bedding; 15 cm long×25 cm wide×30 cm high) connected to a Box 2 (45 × 30 x 30 cm) by a hallway (25×10×30 cm³). The Box 1 was separated from the hallway by a sliding door, and the Box 2 was divided into two compartments (15 cm long and 30 cm long) by a wall, which was removed during the predator exposure test (PET) (see Supplementary material).

### Habituation phase

For 5 days, each mouse was placed into Box 1 and left undisturbed for 5 min. Next, we opened the Box 1 sliding door, and the animal was allowed to explore the hallway and the Box 2 for 10 min. At the end of this session, the mouse was returned to its home cage.

### Predator exposure test

On the sixth day, a neutered 2-year-old male cat was placed and held in the Box 2 by an experimenter, and the mouse was placed into Boxes 1 and 5 min after, the Box 1 sliding door was opened, and the animals were exposed for 5 min to the cat. After the cat was removed at the end of the 5-min period, the mouse was placed back into its home cage and the hallway and Box 2 were then cleaned with 5% alcohol and dried with paper towels.

### Context

On the day after the cat exposure, the mouse was placed back into Boxes 1 and 5 min after, the Box 1 sliding door was opened, and the mouse was exposed for 5 min to the environment where the predator had been previously encountered.

Note that the experimenter was present in the experimental room during the habituation phase, PET, and Context. In all conditions, the experimenter's position in the experimental room remained consistent.

### Behavior analysis

All behavioral sessions were recorded using a high-speed (120 fps) camera (DMC-FZ200, Panasonic) and they were blindly scored by a trained observer using the BORIS software (Behavior Observation Research Interactive Software). The behavioral data were processed in terms of duration (total duration per session). The following behaviors were measured:

- Freezing: cessation of all movements, except for those associated with breathing;
- Risk-assessment behaviors: comprising crouch-sniff (animal immobile with the back arched, but actively sniffing and scanning the environment) and stretch postures (consisting of both stretch attend posture, during which the body is stretched forward and the animal is motionless, and stretch approach, consisting of movement directed toward the cat compartment with the animal's body in a stretched position);
- Fearless exploration: including nondefensive locomotion and exploratory up-right position (i.e., animals actively exploring the environment, standing over the rear paws, and leaning on the walls with the forepaws).

## Pharmacogenetic inhibition

Animals were previously habituated to the handling, and on the last 3 days of the habituation phase received injections of 0.2 ml of saline i.p. For the pharmacogenetic inhibition, animals were injected with 1 mg/kg of clozapine-N-oxide i.p. (CNO; Tocris Bioscience, UK) 30 min before the beginning of the behavioral test.

### Experimental design

Animals were tested for the Acquisition and the Expression phases. For the Acquisition phase, we tested a group of animals expressing Gi-coupled hM4Di (hM4D+; n=7) and a control group (hM4D−; n=6), both of which were injected with CNO prior to the PET and saline prior to the Context (*Figure 1B*). Similarly, for the Expression phase, we tested a group of animals expressing Gi-coupled hM4Di (hM4D+; n=7) and a control group (hM4D−; n=5), both of which received saline injection prior to the PET and CNO injection prior to the Context exposure (*Figure 1C*). For the pharmacogenetic inhibition, the criteria for including the experiments were based on the correct position of viral transfection.

## Optogenetic inhibition

Animals were previously habituated to the optogenetic cables on the last 3 days of the habituation phase of the behavioral paradigm. This procedure consisted of plugging optogenetic cables to the implanted fiber-optic cannulae and letting the animals explore the apparatus for 5 min without any additional stimulus. Optogenetic inhibition was induced by exposing animals continuously to a yellow laser (589 nm, Low-Noise DPSS Laser System, Laserglow Techonologies). Note that the continuous yellow laser stimulation did not produce tissue damage in the soundings of the optic fiber tips (see *Appendix 1—figure 11*).

Experimental design. For the photoinhibition of the AM>ACA pathway, we tested a group of animals expressing halorhodopsin-3.0 (HR+; n=8) and a control group (HR−; n=7), both of which had the yellow laser turned ON during PET and turned OFF during the context exposure. For the photoinhibition of the ACA projections to the selected targets during acquisition and expression of contextual fear responses, animals were tested for the Acquisition and the Expression phases. For the Acquisition phase, we tested groups of animals expressing halorhodopsin-3.0 (HR+) and control groups (HR−) that had the yellow laser turned ON during PET and turned OFF during the context exposure (ACA>BLA pathway – HR+ n=8, HR− n=6; ACA>PERI pathway – HR+ n=6, HR− n=5; ACA>POST pathway – HR+ n=6, HR− n=5; ACA>PAGdl pathway – HR+ n=7, HR− n=5). Similarly, for the Expression phase, we tested groups of animals expressing halorhodopsin-3.0 (HR+) and control groups (HR−) that had the yellow laser turned OFF during PET and turned ON during the context exposure (ACA>BLA pathway – HR+ n=8, HR− n=6); ACA>PERI pathway – HR+ n= 6, HR− n=5; ACA>POST pathway – HR+ n=7, HR− n=5; ACA>PAGdl pathway – HR+ n=9, HR− n=5. For optogenetic inhibition, the criteria for including the experiments were based on the correct position of viral transfection and optic fiber placement.

## Functional tracing

### Fluoro Gold injection

FG injections followed the same anesthetic and stereotaxic procedures described for the viral injections. Iontophoretic deposits of 2% solution of Fluoro Gold (FG; Fluorochrome Inc, CO) was applied unilaterally into the ACA (AP +1.0, ML +0.3, DV −1.1), BLA (AP −0.95, ML +3.3, DV −3.45), PERI (AP −3.9, ML −0.4 from the lateral skull surface, DV −0.9 from the local brain surface), POST (at a 20° angle from the vertical axis; AP −3.9, ML +2.7, DV −1.15), or PAGdl (AP −3.85, ML +0.45, DV −2.1). Deposits were made over 5 min through a glass micropipette (tip diameter, 15 μm) by applying a +3 μA current, pulsed at 7-s intervals, with a constant-current source (Midgard Electronics, Wood Dale, IL, model CS3).

### Experimental design

After 7 days of recovering from surgery, we started the behavioral procedures. To study the activation pattern of the different ACA source of inputs during the cat exposure and the predatory context, ACA FG injected animals were perfused 90 min after the PET (n=6) or the Context exposure (n=6). To quantify the relative amount of activation of the ACA projections to the selected targets during the acquisition and expression of contextual fear to predator threat, FG injected animals into the selected targets were perfused 90 min after the PET (BLA n=4; PERI n=4; POST n=4; PAGdl n=4) or the Context exposure (BLA n=4; PERI n=4; POST n=4; PAGdl n=4).

## Triple retrograde tacing

These experiments were conducted following the same anesthetic and stereotaxic procedures previously described. In four animals, we placed on the same side of the brain injections of three different

retrograde tracers, namely, the retrograde AAV-Retro-Ef1a-mcherry-IRES-Cre, Fluoro Gold, and Cholera toxin B (CTb) in the PAGdl, BLA, and PERI, respectively.

### AAV injections
The retrograde AAV-Retro-Ef1a-mcherry- IRES-Cre (titer≥7×10$^{12}$ vg/ml; Karl Deisseroth; Addgene catalog #55632-AAVrg) was injected unilaterally into the PAGdl (AP –3.85, ML +0.45, DV –2.1). Deposits were made with a 5-µl Hamilton Syringe (Neuros Model 7000.5 KH), and a total of 50 nl of virus were delivered at a rate of 5 nl/min. After 2 weeks, we proceeded with the iontophoretic injections of Fluoro Gold into the BLA and CTb into the PERI.

### Fluoro Gold and CTb injections
Iontophoretic deposit of 2% solution of Fluoro Gold (FG; Fluorochrome Inc, CO) was applied unilaterally into the BLA (AP –0.95, ML +3.3, DV –3.45), and a deposit of Cholera toxin B (CTb; 1%; List Biological Laboratories, Campbell, CA) was applied unilaterally into the PERI (AP –3.9, ML –0.4 from the lateral skull surface, DV –0.9 from the local brain surface). Both deposits were made over 5 min through a glass micropipette (tip diameter, 15 µm) by applying a +3 µA current, pulsed at 7-s intervals, with a constant-current source (Midgard Electronics, Wood Dale, IL, model CS3). After 2 weeks, the animals were perfused, and we performed the histological processing.

## Perfusion and histological processing
After the experimental procedures, animals were deeply anesthetized in a box saturated with Isoforine and transcardially perfused with a solution of 4.0% paraformaldehyde in 0.1 M phosphate buffer at pH 7.4. The brains were removed and left overnight in a solution of 20% sucrose in 0.1 M phosphate buffer at 4°C. The brains were then frozen, and five series of 30-mm-thick sections were cut with a sliding microtome in the frontal plane. Sections from all the viruses and FG injections were taken to the fluorescent microscope to verify the injection sites and projection fields evidenced by the fluorescent reporters.

### Functional tracing – FG/Fos immunofluorescence double-labeling
One series of sections was washed with KPBS to remove the cryoprotectant solution and then incubated with polyclonal rabbit anti-c-Fos (PC-38; Calbiochem-Millipore) at 1:20,000 in 0.5% Blocking Reagent (Roche) in KPBS for 48 hr at 4°C. Sections were then washed in KPBS and incubated with Anti-Rabbit Alexa 594 Goat IgG (H+L) (Invitrogen) at 1:500 during 2 hr at random temperature (RT). Sections were then washed in KPBS and incubated with rabbit anti-FG antibody (1:5000; Chemicon International, CA) in 0.5% Blocking Reagent (Roche) in KPBS for 16 hr at 4°C. The Sections were then washed in KPBS and incubated with Anti-Rabbit Alexa 488 Goat IgG (H+L) (Invitrogen) at 1:1000 during 1.5 hr at RT. At the end, the sections were mounted in gelatin-coated slides, the nuclei were counterstained with DAPI (Sigma-Aldrich) at 1:20,000 in TBS and cover slipped with Fluoromount (Sigma-Aldrich). Slides were stored at 4°C in humid chambers until image capture.

### Triple retrograde tracing – FG and CTb immunofluorescence
Both immunofluorescences were done at the same essay. The immunofluorescence for FG was prior described, but in this experiment, we used the anti-rabbit Alexa 405 goat IgG (H+L) at 1:500 as secondary antibody. For the CTb immunofluorescence, the sections were incubated with antisera against CTb (raised in goat, 1:20,000; List Biological Laboratories,Campbell, CA) overnight at RT. Then the sections were washed in KPBS and incubated with anti-goat Alexa 488 donkey IgG (H+L) (Invitrogen) at 1:500 during 1.5 hr at RT. After the histological processing, we captured the images and analyzed the retrogradely labeled neurons in the ACA.

## Image capture and analysis
Brain regions of interest were determined using the *Allen Mouse Brain Atlas*. Images were captured using an epi-fluorescence microscope (NIKON, Eclipse E400) coupled to a digital camera (NIKON, DMX 1200). ImageJ public domain image processing software (FIJI v1.47f) was used for image analysis. For the documentation of the viral injection sites and projection fields of viral transfected cells, we captured only the fluorescence emitted from the fluorescent reporters. For the functional tracing

studies, Fos and FG were immunodetected and labeled with Alexa 594 and Alexa 488, respectively. To determine the activation pattern of the different ACA source of inputs during the cat exposure and the predatory context, we first selected the sites containing more than 10% of FG-labeled cells expressing Fos protein. For each one of these sites, we first delineated the borders of the selected region and FG and FG/FOS double-labeled cells were counted therein. The number of counted cells were corrected by the quantified area. To quantify the relative amount of activation of the ACA projections to the selected targets during the acquisition and expression of contextual fear to predator threat, we averaged the counting from three serial 60 μm apart sections, where, in each section, we delineated the ACA borders, counted FG, FOS, FG/FOS labeled cells as well as DAPI stained cells and corrected by the quantified area.

## Electrophysiology

### Slice preparation

Slices were prepared using a Leica VT1000s vibratome as described previously (*McKay et al., 2009*; *Oh et al., 2013*). Briefly, mice were deeply anesthetized with isoflurane and decapitated. The brains were quickly removed, immersed in ice-cold aCSF (artificial cerebrospinal fluid) solution bubbled with 95%$O_2$/5%$CO_2$ and sliced. The aCSF solution containing (in mM): 125 NaCl, 2.5 KCl, 1.25 $NaH_2PO_4$, 26 $NaHCO_3$, 2 $CaCl_2$, 1 $MgSO_4$, and 25 glucose. The cut slices were transferred immediately to a warm (37°C) submerged holding chamber filled with aCSF and kept in this chamber for 30 min. Next, the slices were allowed to return to RT (~25°C) for at least 40 min before electrophysiology experiments.

### Patch-clamp

Whole-cell voltage or current-clamp recordings were made in aCSF at 34±0.5°C from visualized neurons using a CMOS camera (Flash4.0, Hamamatsu) mounted on SliceScope Pro 3000 microscope (Scientifica, UK); using a long working distance 40× (0.8 NA) water-immersion objective and infrared differential interference contrast optics. For current-clamp experiments, patch electrodes, yielding 4–6 MΩ resistance, contained (in mM): 130 $KMeSO_4$, 10 KCl, 10 HEPES, 4 ATP magnesium salt, 0.4 GTP disodium salt, with pH corrected to 7.3 with KOH and osmolarity of 295±5 mOsm. Neurons were included if they had a resting membrane potential of less than –60 mV, an input resistance >25 MΩ, AP amplitude of >80 mV from rest (calculated from the resting potential of the cell until the peak of the AP, which was evoked by brief square pulses [2 ms, 1.5 nA]), and stable series resistance of <20 MΩ. Resting membrane potential was measured immediately after breaking into the cell. Electrode capacitance and series resistance were monitored and compensated throughout recording; cells were held between –65 and –66 mV with injected current. Input resistance was calculated as the slope of the V-I curve using 500 ms current steps from ⩾300 to 0 pA at 50 pA steps. APs were evoked using ramp current pulses (100 ms pulses ranging from 100 to 500 pA at 100 pA steps). For voltage-clamp recordings, patch electrodes, yielding 4–6 MΩ resistance, contained (in mM): 130 CsCl, 5 EGTA, 10 HEPES, 4 ATP magnesium salt, 0.4 GTP disodium salt, 10 TEA, 6 QX314, and 5 Creatine phosphate disodium salt with pH corrected to 7.3 using CsOH and osmolarity of 295±5 mOsm. Electrical stimulation was delivered as a square pulse of 100 μs duration with an intensity of 100 μA using a concentric bipolar electrode (30201, FHC, INC). After breaking the whole-cell configuration, the cells were held at –60 by current injection and the stimulating electrode was moved until EPSCs were detected in the postsynaptic cell. Stimuli were delivered every 1 s and the stimulating electrode was between 50 and 100 μm far from the cells recorded. Data were collected using a Multiclamp 700A amplifier, pClamp 10.7 software, and digitized (10 kHz) using a Digidata 1440 AD converter. All from Molecular Devices (USA). Data were analyzed using Clampfit 10.7 software (Molecular Devices, USA).

### Optogenetics

The transfected neurons were identified and stimulated using a pE-2 light source (CoolLED, UK). The wavelength of 470 nm was used to identify the transfected cells and the wavelength of 585 nm was used to activate halorhodopsin. Both wavelengths were filtered using a double-band fluorescence cube (59022-filter set; Chroma, USA).

## Pharmacogenetics

The neurons transfected with the pAAV-hSyn-HA-hM4D(Gi)-IRES-mCitrine or AAV5-hSyn-eGFP were identified using a pE-2 light source (CoolLED, UK). The wavelength of 470 nm, filtered by the 59022 cube-set (Chroma, USA) was used to identify the transfected cells in the brain slices. CNO was superfused at a flow rate of ~3 ml/min.

## Statistical analysis for FG/Fos double-immunofluorescence

To evaluate the potential discrepancy between the proportion of FG/Fos double-labeled cells in the cat exposure condition in comparison to the predatory context condition, we employed a 2×2 Chi-square test with Yates' correction for continuity, separately for each injection site. To maintain the overall type I error at 5%, the significance level employed in each test was adjusted downward (Bonferroni's correction) according to the total number of analyses performed ($\alpha=0.0125$). The 95% CIs for proportions were calculated using ESCI – Exploratory Software for Confidence Intervals (*Cumming, 2012*).

The observed FG/Fos immunofluorescence double-labeling was compared to chance-level overlap by calculating a double-to-chance ratio as follows: *double labeling*=(number of FG/Fos double-labeled cells)/(total number of DAPI-positive cells); *chance level*=(number of FG-labeled cells/total number of DAPI-positive cells)×(number of Fos-labeled cells/total number of DAPI-positive cells); *double-to-chance ratio=double labeling/chance level* (*Yokoyama and Matsuo, 2016*). A ratio close to unity means that the observed double-labeling count does not differ from the chance level, whereas a ratio significantly distant from 1 (gauged by a CI) indicates an above chance double labeling. The ratios were calculated individually and averaged over the animals in each group (N=4) separately, allowing the calculation of t-scores 95% CIs (a CI spanning unity indicates a non-significant double-to-chance ratio, the opposite being suggested by CIs that exclude it).

## Statistical analysis for behavioral measurements

After testing for homogeneity of variance (Levine's test), the behavioral data were square-root transformed whenever the null hypothesis of homoscedasticity was rejected. For all experiments, the analysis was performed by means of a parametric univariate analysis of variance (ANOVA), followed by a post hoc analysis (Tukey's HSD test) when appropriate. The specific design varied depending on the structure of each experiment, ranging from one-way to three-way ANOVAs. Due to an expressive number of analyses performed and to maintain the overall type I error at 5%, the significance level employed in each ANOVA was adjusted downward by means of a Bonferroni's correction ($\alpha=0.0028$). The average results are expressed as the mean ± SEM throughout the text and effect sizes are expressed as partial eta-squared. Two-tailed tests were used throughout the statistical analyses for both cell counting and behavioral measurements. Note that the statistical analysis was conducted by an experimenter (MVCB) without previous knowledge of the experimental results.

## Acknowledgements

This research was supported by Fundação de Amparo à Pesquisa do Estado de São Paulo (FAPESP) Research Grants #2014/05432-9 (to NSC) and #2019/27245-0 (to FAO). FAPESP fellowships to MAXL (#2016/10389-0).

## Additional information

### Funding

| Funder | Grant reference number | Author |
|---|---|---|
| Fundação de Amparo à Pesquisa do Estado de São Paulo | #2014/05432-9 | Newton Sabino Canteras |
| Fundação de Amparo à Pesquisa do Estado de São Paulo | #2016/10389-0 | Miguel Antonio Xavier de Lima |

| Funder | Grant reference number | Author |
|---|---|---|
| Fundação de Amparo à Pesquisa do Estado de São Paulo | #2019/27245-0 | Fernando A Oliveira |

The funders had no role in study design, data collection and interpretation, or the decision to submit the work for publication.

## Author contributions

Miguel Antonio Xavier de Lima, Conceptualization, Data curation, Formal analysis, Investigation, Methodology, Writing – original draft; Marcus Vinicius C Baldo, Formal analysis, Statiscal analysis, Supervision; Fernando A Oliveira, Data curation, Formal analysis, Methodology, performed and analyzed the patch clamp studies; Newton Sabino Canteras, Conceptualization, Formal analysis, Funding acquisition, Supervision, Writing – original draft, Writing – review and editing

## Author ORCIDs

Fernando A Oliveira http://orcid.org/0000-0002-1632-4267
Newton Sabino Canteras http://orcid.org/0000-0002-7205-5372

## Ethics

All experiments and conditions of animal housing were carried out under institutional guidelines [Colégio Brasileiro de Experimentação Animal (COBEA)] and were in accordance with the NIH Guide for the Care and Use of Laboratory Animals (NIH Publications No. 80-23, 1996). All of the experimental procedures had been previously approved by the Committee on the Care and Use of Laboratory Animals of the Institute of Biomedical Sciences, University of Sã Paulo, Brazil (Protocol No. 085/2012). Experiments were always planned to minimize the number of animals used and their suffering.

## Decision letter and Author response

Decision letter https://doi.org/10.7554/eLife.67007.sa1
Author response https://doi.org/10.7554/eLife.67007.sa2

# Additional files

## Supplementary files

• Transparent reporting form

## Data availability

All data generated and analysed in this study are included in the manuscript and supporting files.

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

# Appendix 1

## Behavioral protocol

### Exposure to the predator and predatory context

In the present study, mice were exposed to a predator (a live cat) and tested for innate and contextual fear responses. Animals were tested in an experimental apparatus that consisted of a 25×15×30 cm$^3$ home cage (Box 1) connected to another 30×45×30 cm$^3$ chamber (Box 2) by a hallway that was 10 cm wide, 25 cm long, and 30 cm high (*Appendix 1—figure 1*). As shown in *Appendix 1—figure 2A*, during 5 days before testing, animals were habituated to the apparatus. The relatively long habituation period seems necessary to optimally stabilize both innate and contextual defensive responses (*Ribeiro-Barbosa et al., 2005*). On the sixth day, a neutered 2-year-old male cat was placed and held in the Box 2 by an experimenter, and the mouse was placed into Boxes 1 and 5 min after, the Box 1 sliding door was opened, and the animals were exposed for 5 min to the cat (*Appendix 1—figure 2B*, Predator Exposure Test – PET Condition). Male and female mice exposed to the cat presented clear innate defensive responses, they froze for approximately 30% of the time and presented clear risk assessment responses, characterized by crouch sniff, and stretch postures toward the cat, close to 30% of the time (*Appendix 1—table 1*). After the cat was removed at the end of the 5-min period, the mouse was placed back into its home cage and the hallway and Box 2 were then cleaned with 5% alcohol and dried with paper towels. On the following day, animals were exposed to the same environment of the cat exposure during 5 min (*Appendix 1—figure 2C*, Context Condition) and, presented risk assessment responses close to 50% of the observational period, but no freezing (*Appendix 1—table 1*). Similar to previous results (*Ribeiro-Barbosa et al., 2005*), contextual fear responses to a predator threat are characterized mostly by risk assessment behaviors. In the case of mice, we observed only risk assessment with no freezing, and in rats exposed to the predatory context, it has been reported a great deal of risk assessment with a reduced amount of freezing (*Ribeiro-Barbosa et al., 2005*). In the present experiments, mice presented a few escape episodes during cat exposure, which occurred mostly when the mice became aware of the cat and fled back to Box 1, and only occasional escape responses during exposure to the predatory context (*Appendix 1—table 2*). An important feature of the experimental procedure used here was that the behavioral responses were very stable among the animals tested within each individual phase of the testing schedule. This feature may be, at least in part, accounted for by the long habituation period, which appears to stabilize anti-predatory behavioral responses (*Ribeiro-Barbosa et al., 2005*).

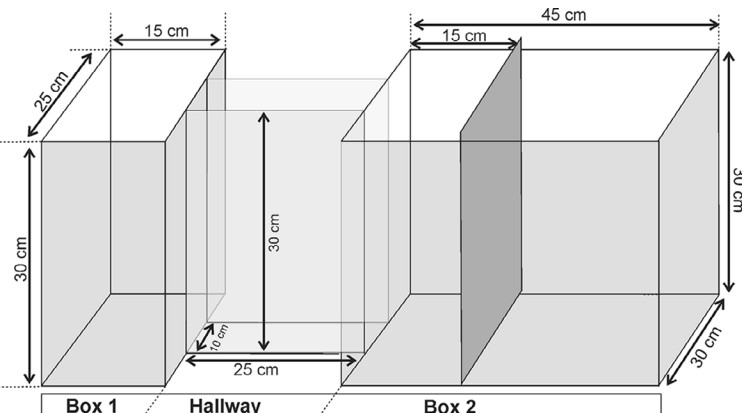

**Appendix 1—figure 1.** Experimental apparatus used for exposure to the Predator and Predatory Context.

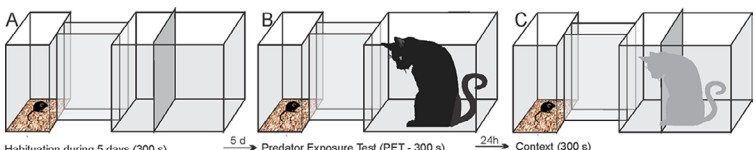

**Appendix 1—figure 2.** Timeline to illustrate the experimental procedure.

**Appendix 1—table 1.** Behavioral data from male (n=6) and female (n=6) mice during the PET and Context conditions.

|  | Predator exposure test (PET) | | Context | |
|---|---|---|---|---|
|  | Male | Female | Male | Female |
| *Freezing* | 102.8±4.1 | 98.2±5.5 | 0±0 | 0±0 |
| *Risk assessment* | 107.6±3.5 | 110.9±6.2 | 161.6±8.2 | 162.8±6.9 |
| *Exploration* | 25.8±2.5 | 31.9±3.0 | 37.9±4.4 | 34.7±5.2 |

**Appendix 1—table 2.** Escape responses.

ACA – Pharmaco inhibition

| Phase | GROUP | *PET_Esc* | *CONT_Esc* |
|---|---|---|---|
| *Acquisition* | hM4D+ | 1.57±0.3 | 0.43±0.2 |
| *Acquisition* | hM4D− | 1.67±0.21 | 0.33±0.33 |
| *Expression* | hM4D+ | 1.57±0.3 | 0.57± 0.2 |
| *Expression* | hM4D− | 1.6±0.4 | 0.4±0.24 |

AM>ACA Opto inhibition

| Phase | GROUP | *PET_Esc* | *Cont_Esc* |
|---|---|---|---|
| *Acquisition* | HR+ | 1.5±0.28 | 0.5±0.26 |
| *Acquisition* | HR− | 1.43±0.3 | 0.43±0.2 |

ACA>BLA Opto inhibition

| Phase | GROUP | *PET_Esc* | *CONT_Esc* |
|---|---|---|---|
| *Acquisition* | HR+ | 1.5±0.26 | 0.37±0.18 |
| *Acquisition* | HR− | 1.33±0.21 | 0.33±0.21 |
| *Expression* | HR+ | 1.28±0.18 | 0.5±0.27 |
| *Expression* | HR− | 1.5±0.34 | 0.33±0.21 |

ACA>PERI Opto inhibition

| Phase | GROUP | *PET_Esc* | *CONT_Esc* |
|---|---|---|---|
| *Acquisition* | HR+ | 1.66±0.33 | 0.16±0.16 |
| *Acquisition* | HR− | 1.4± 0.24 | 0.4±0.24 |
| *Expression* | HR+ | 1.5±0.22 | 0.16±0.16 |
| *Expression* | HR− | 1±0 | 0.2±0.19 |

ACA>POST Opto inhibition

| Phase | GROUP | *PET_Esc* | *CONT_Esc* |
|---|---|---|---|
| *Acquisition* | HR+ | 1.33±0.21 | 0±0 |
| *Acquisition* | HR− | 1.2±0.37 | 0.4±0.24 |
| *Expression* | HR+ | 1.16±0.18 | 0.33±0.18 |
| *Expression* | HR− | 1.2±0.19 | 0±0 |

ACA>PAG Opto inihibition

| Phase | GROUP | *PET_Esc* | *CONT_Esc* |
|---|---|---|---|
| *Acquisition* | HR+ | 1.43±0.14 | 0.14±0.14 |
| *Acquisition* | HR− | 1.4±0.24 | 0.2±0.22 |
| *Expression* | HR+ | 1.5±0.24 | 0.33±0.23 |
| *Expression* | HR− | 1±0 | 0.2±0.19 |

Time in seconds (mean ± SEM) spent by both groups of rats (male and female) when expressing different behaviors (freezing, risk assessment, and exploration) under two experimental conditions (PET and context). A univariate 2×2 factorial ANOVA, with the between-subjects factor Sex (Male and Female) and the within-subjects factor Exposition (PET and Context), revealed a significant main effect of the Exposition factor on both freezing and risk assessment behaviors (Freezing: $F_{1,10}$= 868.11, p<0.001, $\eta^2$=0.976; Risk assessment: $F_{1,10}$=80.19, p<0.0001, $\eta^2$=0.768) and a marginally significant effect on the exploration behavior ($F_{1,10}$=3.63, p=0.086, $\eta^2$=0.143). No statistically significant effect of the factor Sex on any of the behaviors was observed (Freezing: $F_{1,10}$=0.46, p=0.515, $\eta^2$≤0.001; Risk assessment: $F_{1,10}$=0.11, p=0.751, $\eta^2$=0.001; Exploration: $F_{1,10}$=0.14, p=0.721, $\eta^2$=0.005); also, no significant interaction between the factors Sex and Exposition was found for any of the behaviors (Freezing: $F_{1,10}$=0.46, p=0.515, $\eta^2$<0.001; Risk assessment: $F_{1,10}$=0.03, p=0.862, $\eta^2$<0.001; Exploration: $F_{1,10}$=1.41, p=0.262, $\eta^2$=0.056).

Number of escape episodes during cat (PET) and context (CONT) exposure. We listed for the experimental (hM4D+ and HR+) and control (hM4D− and HR−) groups, all behavioral experiments with chemo- or opto-inhibition during the different phases of fear memory (acquisition and expression). Values expressed as mean ± SEM.

**Appendix 1—video 1.** Exposure to the predator and predatory context. Video to illustrate the behavioral responses of a male mice (C57BL/6) exposed to the cat (PET condition) and the predator-associated environment (Context condition).

https://elifesciences.org/articles/67007/figures#video1

## Comparison of ACA activity
### Comparison of ACA activity in response to cat and novel non-threat stimulus (plush cat)

In the ACA, DAPI, and Fos quantification were performed in four conditions: exposure to the cat (n=6), the predatory context (n=6), the novel non-threat stimulus (plush cat) (n=6), and the context paired with non-threat stimulus (n=6). The procedure to cat and plush cat exposure, as well as exposures to their related contexts, was identical to that described previously for the predator exposure test (PET condition) and context condition.

The comparison between the activities of the ACA under the experimental (cat exposure vs. predatory context) and control (plush cat exposure vs. plush cat-related context) conditions was carried out by means of a 2× 2 two-way between-subject ANOVA, with the factors Predator (real cat vs. plush cat) and Condition (predator exposure vs. context exposure) as independent variables, and the fraction Fos+/DAPI for each animal as the dependent variable. The normality of the data and the homogeneity of variances were checked by means of Shapiro-Wilk tests combined with Q-Q plots and Levene's test, respectively. The Tukey's HSD test was employed to run post hoc comparisons after obtaining statistically significant effects in the omnibus ANOVA.

After checking for normality and homogeneity of variances (the sample in the present experiment passed both tests) a 2×2 two-way between-subject ANOVA revealed significant main effects for both factors, Predator and Condition (respectively, $F_{(1,20)}=284.7$, $p<0.0001$, partial eta squared $=0.934$ and $F_{(1,20)}=68.3$, $p<0.0001$, partial eta squared $=0.773$). In fact, when the plush cat was used as a mock predator, the average fractions of the Fos+/DAPI counting in both conditions (either predator or context exposure) was significantly smaller than the Fos+/DAPI averages found in the corresponding real cat conditions (Tukey's HSD test, $p<0.0001$). A significant difference was found between the effects of predator and context exposures for the plush cat on the ACA activity (Tukey's HSD test, $p=0.025$). However, a statistically significant interaction between Predator and Condition factors ($F_{(1,20)}=14.7$, $p=0.001$, partial eta squared$=0.424$) showed that the exposition of the mice to the predator was even greater in comparison to the context condition when the predator was a real cat instead of a plush cat (Tukey's HSD test, $p<0.0001$) (*Appendix 1—figure 3*).

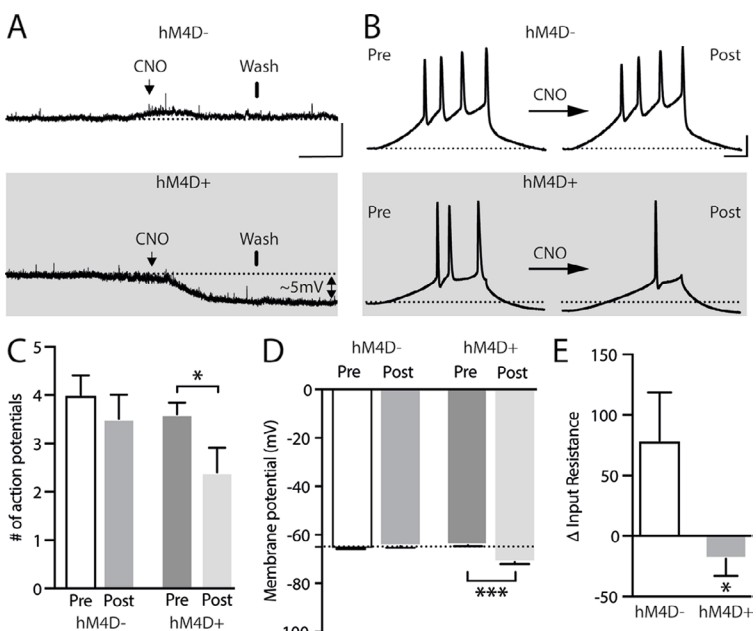

**Appendix 1—figure 3.** Comparison of ACA activity in response to cat and novel non-threat stimulus. On the left, drawings to illustrate the experimental conditions PET (predator exposure test), Context (real cat related context), PCE (plush cat exposure), and PCC (plush cat context). On the right, histogram showing the ACA activity expressed as the Fos+/DAPI fraction in both conditions (Predator exposure and Predatory context) when the predator was either a real cat or a plush cat (error bars are 95% C I.).

## Electrophysiology

**Appendix 1—figure 4.** Impact of CNO on resting membrane potential and neuronal excitability in hM4D+ transfected neurons. (**A**) Representative traces of resting membrane potential from hM4D− neurons (control – upper panel) or hM4D+ neurons (lower panel). hM4D+ neurons hyperpolarized as they underwent extracellular CNO. The arrows show the application of the CNO (lasting around 3 min) and the bars show the start of drug washing. Scale bar for panel (**A**): 10 mV, 1.5 min. (**B**): A ramp protocol (100 ms pulse of 500 pA) was used to assess cellular excitability after applying CNO. Representative traces of APs evoked during the ramp protocol in hM4D− and hM4D+ neurons before and after CNO application. (**C**) Bar graph shows a significant decrease in the triggering of APs after 10 µM CNO in hM4D+ neurons (pre: 3.6±0.2 and post 2.4±0.5; n=5) when compared to hM4D− neurons (pre: 4±0.4 and post 3.5±0.5; n=5). (**D, E**) Quantification of the reduction in the resting membrane potential (hM4D−: pre: −65.8±0.14 and post: −64.7±0.6; n=5; hM4D+: pre: −66.1±0.4 and post: −71.3±0.8; n=6; ***p<0.001, paired t-test) and input resistance (hM4D−: 78.8±40; n=5; hM4D+: −18.8±14.2; n=5; *p<0.05, unpaired t-test) caused by the application of CNO. The input resistance was calculated as the slope of the I-V curve and quantified before and after the application of CNO, the difference ($\Delta = R_{final} - R_{initial}$) being reported in the graph. Bars represent mean ± SEM. The dotted line on panels (**A**), (**B**), and (**D**) represents −65 mV.

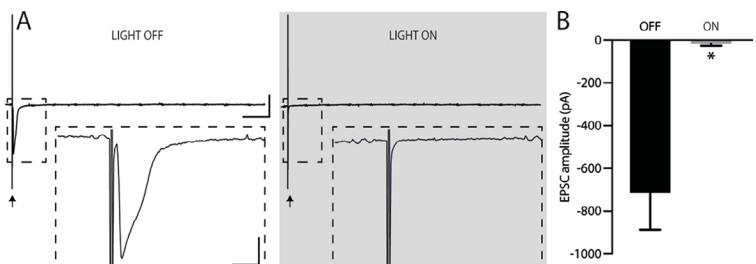

**Appendix 1—figure 5.** Light-induced hyperpolarization in Halorhodopsin positive neurons (HR+). (**A**) Representative trace of membrane potential from an HR+ neuron. The arrows and bars represent the light on and off, respectively. The percentage on the top of the panel indicates the intensity of the light which corresponds to 12.8 (5%), 25 (10%), and 50.6 (20%) μW. Scale bar for panel (**A**): 10 mV, 0.5 min. (**B**): Membrane potential recorded during 585 nm-lights on in the control (HR−) (Basal: –66.4±0.8; 5%: –64.9±0.2; 10%: –65.27±0.3; and 20%: –66.6±0.8; n=5) and HR+ neurons (Basal: –66.8±0.2; 5%: –83.5±2.5; 10%: –96.1±4.6; and 20%: –105.4±4; n=5, **p<0.01). Hyperpolarization was quantified in the last 100 ms before turning off the light. Bars represent mean ± SEM. Dotted line represents –65 mV. (**C**) Linear regression of hyperpolarization caused by halorhodopsin activation due to different light intensities (HR−: slope = –0.001; HR+: slope = –2.241. Using a holding potential of –66 mV as a constraint). Points represent mean ± SEM.

**Appendix 1—figure 6.** Efficiency of pre-synaptic inhibition. (**A**) Representative EPSC traces before and after illumination at 585 nm (25 μW) with the arrows marking the stimulation (stimulus artifact). The upper traces show that there was no rebound excitation after halorhodopsin activation. Bars: 500 pA; 20 ms. The inserts illustrate the reduction in postsynaptic EPSC amplitude during presynaptic halorhodopsin activation. Bars: 200 pA; 2 ms. Illumination started 50 ms before and ended 50 ms after electrical stimulation (*Kaneda et al., 2011*; *Mahn et al., 2016*). (**B**) Quantification of the EPSC amplitude reduction due to halorhodopsin activation in the presynaptic neuron (Light off: –713.2±174.1 mV; Light on: –16.3±9.8 mV; n=4, *p<0.05, paired t-test). After activation of presynaptic halorhodopsin, EPSCs showed an 81% reduction in amplitude. Holding potential of –60 mV. Bars represent mean ± SEM.

## ACA projections

Here, we provide an analysis of the ACA projections to the anteromedial visual and retrosplenial areas, as well as the sites presently investigated as putatively involved in the acquisition and/or expression of contextual fear responses to predatory threats, namely the basolateral amygdalar nucleus (BLA), the perirhinal area (PERI), the postsubiculum (POST), and the dorsolateral part of the periaqueductal grey (PAGdl). To this end, mice (n=3) were unilaterally injected into the ACA (AP +1.0, ML ±0.3, DV –1.1) with 80 nl of AAV5-hSyn-eNpHR3-mCherry. All injections were largely confined to the ACA (*Appendix 1—figure 7*) including its dorsal and ventral parts at the level receive the densest projection from the ventral part of the anteromedial thalamic nucleus (AMv). All cases revealed similar pattern of projections along the brain, and we chose the one with the densest terminal fields.

In the BLA complex, the ACA projects densely to the basolateral nucleus, and avoids the lateral nucleus (*Appendix 1—figure 7B*). In the perirhinal area, a dense projection was found in the middle layers (II–V), spreading ventrally, to a lesser degree, to the lateral entorhinal area (*Appendix 1—figure 7E*). In the postsubiculum, the ACA provides a bilaminar pattern of projection aimed at the superficial and deep layers (*Appendix 1—figure 7E*). In the

periaqueductal gray, the ACA provides particularly dense projection fields to the dorsolateral part, extending, to a lesser degree, to the lateral, and dorsomedial parts (*Appendix 1—figure 7D*).

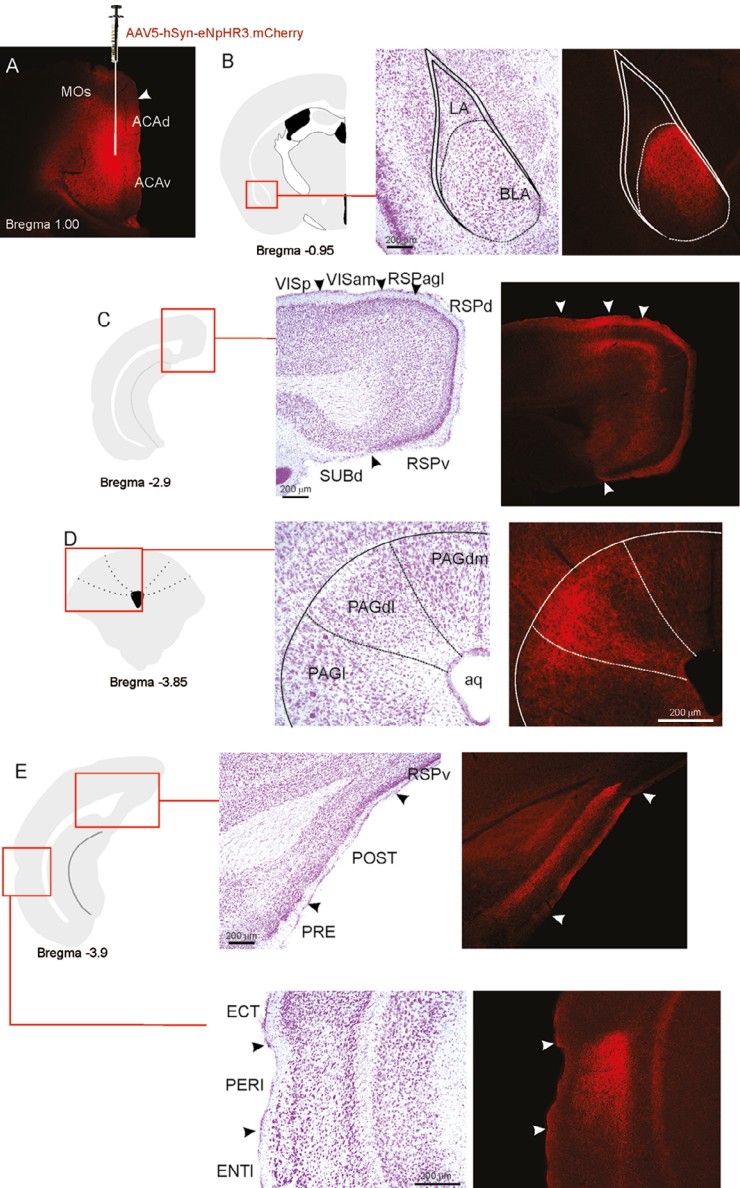

**Appendix 1—figure 7.** ACA projections. (**A**) Fluorescence photomicrograph illustrating the extent and location of unilateral viral injection in the ACA. (**B–E**) On the left, schematic drawings from the *Allen Mouse Brain Atlas* to indicate the locations of the higher magnification nissl-stained and accompanying fluorescence photomicrographs (on the right) to show the ACA projections to the BLA (**B**), anteromedial visual and retrosplenial areas (**C**), PAGdl (**D**), and POST and PERI (**E**). Abbreviations: ACAd, anterior cingulate area, dorsal part; ACAv, anterior cingulate area, ventral part; Aq, aqueduct; BLA, basolateral amygdalar nucleus; ECT, ectorhinal area; ENTl, entorhinal area, lateral part; LA, lateral amygdalar nucleus; MOs, secondary motor area; PAGdl, periaqueductal gray, dorsolateral part; PAGdm, periaqueductal gray, dorsomedial part; PAGl, periaqueductal gray, lateral part; PERI, perirhinal area; POST, postsubiculum; PRE, presubiculum; RSPagl, retrosplenial área, lateral agranular part; RSPd, retrosplenial area, dorsal part; RSPv, retrosplenial area, ventral part; SUBd, subiculum, dorsal part; VISam, anteromedial visual area; VISp, primary visual area.

## Quantification of ACA terminal fields

### Quantification of ACA terminal fields in BLA, POST, PERI, and PAGdl

To perform the quantification of the ACA terminal fields in the BLA, POST, PERI, and PAGdl, we selected three cases (case 70, 71, and 72) with injection of the AAV5-hSyn-eNpHR3.0-mCherry into the ACA. We used the software Fiji (ImageJ) to measure the density of terminals in the ACA projection fields and calculated the corrected total field fluorescence (CTFF = Integrated Density – Area of selected field X mean fluorescence of background readings). To assure the fidelity of this analysis, all the images were captured under the same microscopic parameters and no further adjusts were applied to the images.

In all cases, we obtained similar results and the CTFF revealed the densest ACA projection field in the BLA, followed by the projections to the PERI, PAGdl, and POST, which contained the weakest projection from the ACA (*Appendix 1—figure 8*).

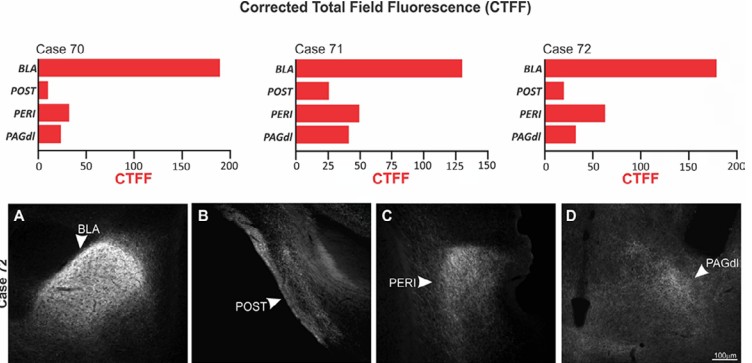

**Appendix 1—figure 8.** Density of ACA terminal fields in BLA, POST, PERI, and PAGdl. *Upper panel*: Corrected total field fluorescence (CTFF) values for the cases 70, 71, and 72 showing the CTFF values for the ACA terminal fields to the BAL, POST, PERI, and PAGdl in a selected case (case 72). *Bottom panel*: Fluorescence photomicrograph illustrating the ACA projection fields in the BLA (**A**), POST (**B**), PERI (**C**), and PAGdl (**D**). Abbreviations: BLA, basolateral amygdalar nucleus; PAGdl, periaqueductal gray, dorsolateral part; PERI, perirhinal area; POST, postsubiculum.

## Triple retrograde tracing

### Triple retrograde tracing to investigate the distribution of ACA neurons projecting to the BLA, PERI, and PAGdl

Considering that the effects on acquisition or expression of contextual defensive responses were found in the ACA projections to the BLA, PERI, and PAGdl, we performed triple retrograde tracing in the same animal to investigate the distribution of ACA neurons projecting to the BLA, PERI, and PAGdl.

As illustrated in *Appendix 1—figure 9*, there is a layering pattern of the ACA neurons projecting to BLA, PERI, and PAGdl. Our analysis revealed that the ACA projecting neurons to PAGdl are located fundamentally in infragranular layers (layers V and VI) (*Appendix 1—figure 9E and G*), and neurons projecting to PERI in the supragranular layers (layers II and III) (*Appendix 1—figure 9F and G*). The ACA neurons projecting to BLA were found mainly in supragranular layers (layers II and III), but we also identified occasional retrogradely labeled cells in layer V (*Appendix 1—figure 9D and G*). The ACA>PAGdl projecting neurons presented no double-labeled cells (*Appendix 1—figure 9G*). Conversely, the ACA cells projecting to the BLA, and the PERI presented 51.66% and 46.98% of double labeling, respectively (*Appendix 1—figure 9G – I*). Thus, suggesting that close to 50% of the ACA cells provided a branched pathway to the BLA and PERI.

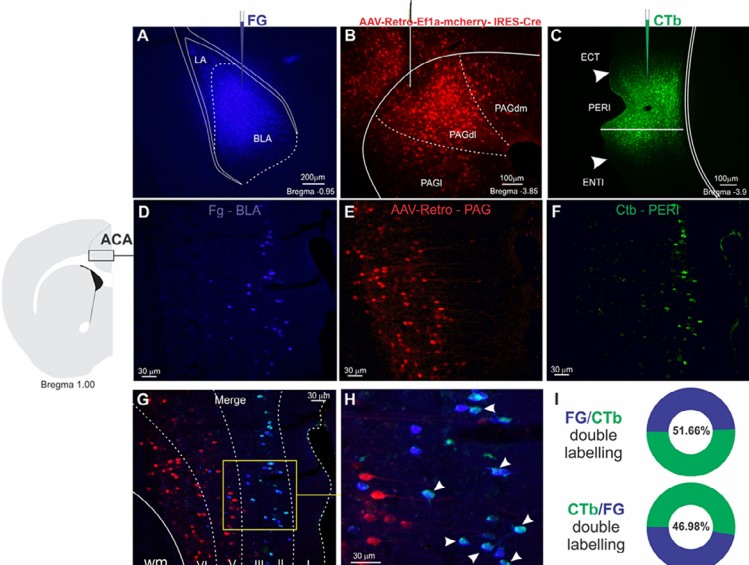

**Appendix 1—figure 9.** Triple retrograde tracing illustrating the distribution of ACA neurons projecting to the BLA, PERI, and PAGdl. *Upper panel*: Fluorescence photomicrographs illustrating the extent and location of unilateral Fluoro Gold deposit in the BLA (**A**), AAV-Retro-Ef1a-mcherry-IRES-Cre deposit in the PAGdl (**B**), and Cholera toxin B deposit in the PERI (**C**). *Intermediate panel*: Schematic drawing from the *Allen Mouse Brain Atlas* to indicate the locations of the ACA (on the left), where the fluorescence photomicrographs illustrate the retrogradely labeled cells from the injections placed in the BLA (**D**), the PAGdl (**E**), and the PERI (**F**). *Bottom panel*: (**G**) fluorescence photomicrograph merging the images shown in (**D**), (**E**), and (**F**) to illustrate the laminar distribution of the ACA retrogradely labeled cells projecting to BLA, PERI, and PAGdl. (**G**) Higher magnification of the insert shown in (**G**) to illustrate double FG-CTb labeled cells, which are indicated with arrowheads. (**I**) Percentage of Fluoro Gold-labeled cells double labeled for CTb (FG/CTb double labeling), and percentage of CTb labeled cells double labeled for Fluoro Gold (CTb/FG double labeling). Abbreviations: ACA: Anterior cingulate area; BLA, basolateral amygdalar nucleus; CTb, Cholera toxin B; ECT, ectorhinal area; ENTl, entorhinal area, lateral part; FG, Fluoro Gold; PAGdl, dm, l, periaqueductal gray, dorsolateral, dorsomedial and lateral parts; PERI, perirhinal area.

## Optic fiber location for optogenetic silencing of the ACA>POST pathway

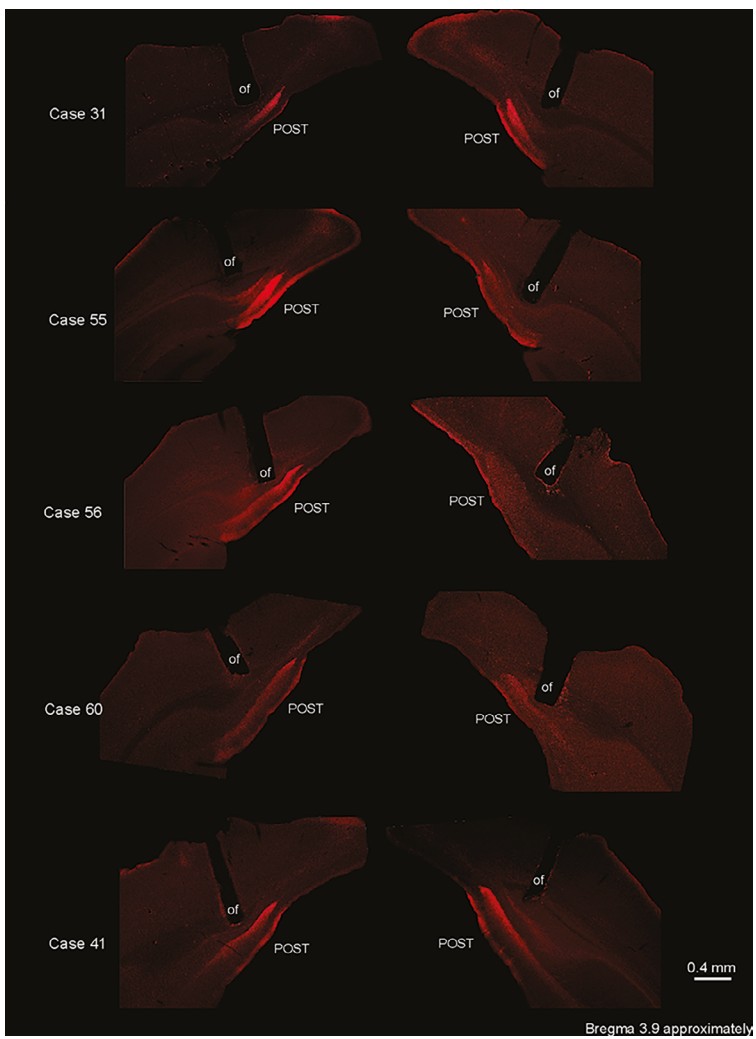

**Appendix 1—figure 10.** Optic fiber location for optogenetic silencing of the ACA>POST pathway. Fluorescence photomicrographs illustrating the optic fiber position in five experiments of the group HR+ that received optogenetic silencing of the ACA>POST pathway during cat exposure. Note that in these experiments, there was a large overlap of the optic fiber's positions, which were suitable to inactivate the ACA>POST pathway. Abbreviations: of, optic fiber; POST, postsubiculum.

## Histological analysis

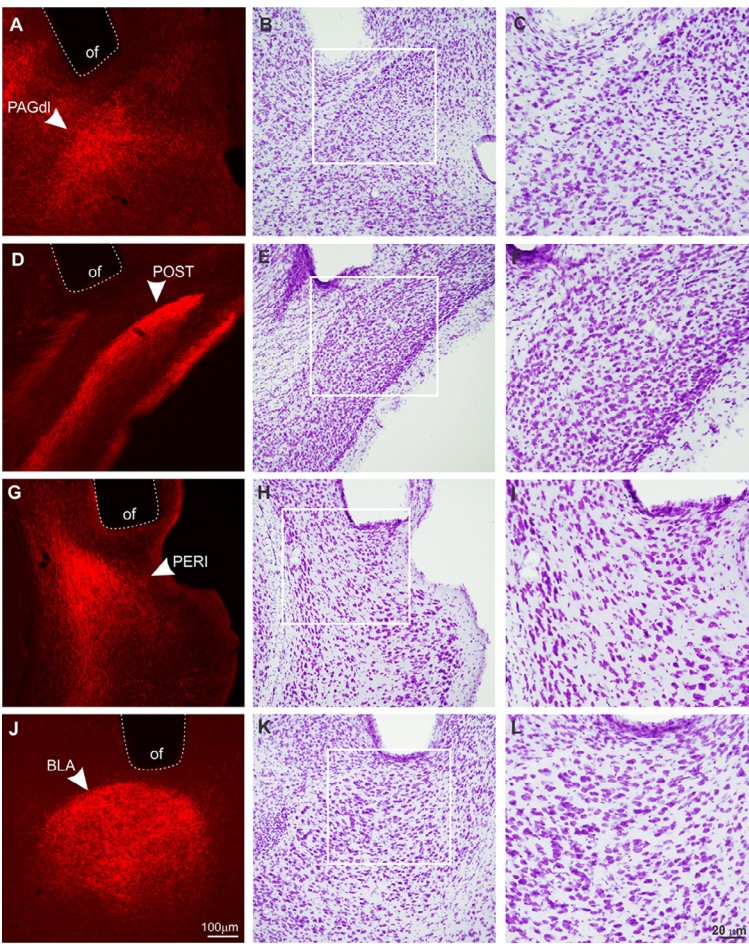

**Appendix 1—figure 11.** Histological analysis on the surroundings of the optic fiber tips. Frontal sections showing the optic fibers (of) tip surroundings at the dorsolateral PAG (PAGdl; **A–C**), postsubiculum (POST; **D–F**), perirhinal area (PERI; **G–I**), and basolateral amygdalar nucleus (BLA; **J–L**). Animals were perfused after behavioral tests. The images shown in (**A**), (**D**), (**G**), and (**J**) illustrate the optic fibers position right above the ACA terminal fields in the PAGdl, POST, PERI, and BLA, respectively. After capturing these images, the same slices were submitted to the Nissl staining protocol (lower magnification – B, E, H, and K; higher magnification – C, F, I, and L). Note that Nissl staining revealed no tissue damage close to the surroundings of the optic fiber tips, evidenced by the lack of gliosis or neuronal loss. Abbreviations: BLA, basolateral amygdalar nucleus; of, optic fiber; PAGdl, periaqueductal gray, dorsolateral part; PERI, perirhinal area; POST, postsubiculum.

