## [Editor Report]

The manuscript provides a detailed circuit analysis of fear learning using a naturalistic task. The identification of anterior cingulate cortex and its input and outputs in contextual fear acquisition and expression to predator threat is an important contribution to our understanding of neural mechanism related to fear processing. The manuscript will be of interest to readers in the field of learning and memory.

---

## [Decision Letter]

**Decision letter after peer review:**

Thank you for submitting your article "The anterior cingulate cortex and its role in controlling contextual fear memory to predatory threats" for consideration by *eLife*. Your article has been reviewed by 3 peer reviewers, one of whom is a member of our Board of Reviewing Editors, and the evaluation has been overseen by Kate Wassum as the Senior Editor. The following individual involved in review of your submission has agreed to reveal their identity: Dayu Lin (Reviewer #3).

Essential revisions:

Methodological. Some concerns over the design were raised, some of which can be addressed through clarifications in the text, but others may require further examination/studies.

1) Inactivation at both time points would help determine if there are state-dependent effects. At minimum this requires discussion, if not additional data.

2) The presence of the experimenter during the acquisition session was a concern shared by the reviewers. It was unclear if the experimenter was present only during acquisition or also during test. If the former, was the position consistent? If the experimenter was not present during test then the authors need to rule out that the presence/absence of the experimenter from acquisition to test did not influence the performance of the mice. This would require additional experiments.

3) Having the laser on for 5 min is long and some evidence that there were no tissue damaging effects would be of value. A control for this stimulation at some timepoint between acquisition and test in the control group is also needed.

4) The efficiency of terminal inhibition using eNpHR3.0 is relatively low. Please use slice recording to confirm that the terminal inhibition can reduce spiking evoked EPSCs of postsynaptic cells and that it does not lead to rebound excitation upon laser off.

5) Additional behavioural measures such as freezing during the test and escape-like responses should be coded and presented.

6) Do the same ACA cells project to BLA, PERI, POST and PAG? It is likely that these projections originate from ACA cells at different cortical layers. The FG tracing experiment should provide some clues. Please use dual retrograde tracing to address the question.

7) For the double FG-Fos double labeled cells, please perform further statistical analysis by comparing the actual overlap with chance level overlap based on Fos% and FG%. This helps understand whether there is a preferential overlap between Fos and FG.

Interpretation.

1) The lack of effect in the ACA>POST pathway inhibition study is taken as evidence that this pathway is not involved. Was the manipulation of this pathway insufficient to obtain an effect; or is it due to the relatively weak projection? The FG+Fos data speak to this somewhat, but those data raise the question with regard to the location of the optical fibers. When there is greater overlap the position of the optical fibers may be less of a concern as it is more likely to target some of those overlapping cells, but this likelihood is decreased in the lower percentage overlap. The histological data may shed light on this issue. Please provide a quantification of the terminal fields in BLA, PERI, POST and PAG.

2) Additional controls are required for the Fos study. These include: Fos expression when the animal is exposed to a non-threat but novel stimulus; Fos expression when the animal is exposed to a novel context or a context paired with a non-threat stimulus. This will help uncover whether the ACA response is specific to threat and threat paired context.

Impact. The lack of female mice is problematic from rigor, translation, and impact perspectives. At minimum please provide a discussion regarding the potential effect of sex on these studies.

[Editors' note: further revisions were suggested prior to acceptance, as described below.]

Thank you for resubmitting your work entitled "The anterior cingulate cortex and its role in controlling contextual fear memory to predatory threats" for further consideration by *eLife*. Your revised article has been reviewed by 3 peer reviewers, one of whom is a member of our Board of Reviewing Editors, and the evaluation has been overseen by Kate Wassum as the Senior Editor.

The manuscript has been improved but there are some remaining issues that need to be addressed, as outlined below:

The reviewers acknowledge that the revised manuscript and the added data constitute a considerable improvement. However, three concerns raised previously were judged to be inadequately addressed. These are outlined below.

1) The response to the state dependent hypothesis is unclear. To clarify the concern, state dependency occurs when information needs to be in the same state at retrieval as during acquisition in order to be recalled. Examples of what may constitute a state include drug influence, emotional states, the activation or inhibition of specific brain area, or pathway etc. This needs to be addressed in the manuscript.

2) The authors offer a descriptive response of the other behaviours observed in their task. The reviewers ask that the authors provide a quantification of those behaviours in the manuscript despite their low frequency.

3) The response to lack of sex-dependent examination was considered insufficient. Since all studies are following up on prior work, there is a logical flaw in using this argument to exclude females because it would prevent females from ever being included in investigations following from work conducted in males. Moreover, the topic investigated here is of high relevance to both males and females and thus inclusion of females is important for the rigor and impact of the work. The authors are asked to provide behavioural data from the task using females. These data must be carefully contrasted to those observed in males and similarities and differences in behaviour and behavioural strategies must be explored and discussed. This is the bare minimum expected. A neurobiological analysis of the effect in females, of course, would strengthen the paper in orders of magnitude. In addition, the term "sex" should be used in place of "gender" because gender is not a construct that can be examined in non-human animals.

4) The response to the ACA-POST projection question regarding whether the pathway is involved or whether it is merely a weak projection does not seem to be addressed in a way that clarifies the concern.

[Editors' note: further revisions were suggested prior to acceptance, as described below.]

Thank you for resubmitting your work entitled "The anterior cingulate cortex and its role in controlling contextual fear memory to predatory threats" for further consideration by *eLife*. Your revised article has been evaluated by Kate Wassum as the Senior Editor, and a Reviewing Editor.

The manuscript has been improved but there are some remaining issues that need to be addressed, as outlined below:

1) The response on state-dependency did not have page numbers and lines that corresponded to appropriate text in the paper or the response in the letter did not align with the wording in the text. This needs to be clarified.

2) Thank you for adding the behavioural data for the females in the supplemental. No neural data were added, which leaves the authors speculating about possible lack of differences at the circuit level based on lack of differences in behaviour. The authors need to acknowledge that a similar behavioural effect does not necessarily mean that the underlying circuits are the same. A more careful discussion of this should be added along with a statement about the need for future investigation into this matter and a more clear acknowledgement of the limitations of the exclusion of females from the neural investigations of this study.

---

## [Author Response]

Essential revisions:Methodological. Some concerns over the design were raised, some of which can be addressed through clarifications in the text, but others may require further examination/studies.1) Inactivation at both time points would help determine if there are state-dependent effects. At minimum this requires discussion, if not additional data.

One of the reviewers raised the possibility of a state-dependent effect on neural manipulation. In order for a given condition to produce a state-dependent effect on memory, the condition should be present during both the acquisition and the expression phases. Therefore, for a given pathway to generate a condition to produce a state-dependent effect on memory, the pathway would be expected to be activated to the same degree during both encoding and retrieval. In this case, manipulation of the activity of the pathway during either encoding or retrieval would impair memory recall. However, in our case, all pathways involved in memory acquisition or expression were differentially activated during each stage and is not expected to generate a state-dependent condition that affects memory. Thus, the pathways that affect memory encoding (i.e., the AM > ACA, ACA > BLA and ACA > PERI pathways) were more active in the encoding rather than retrieval phase. Conversely, the pathway that influenced memory expression (i.e., the ACA > PAG pathway), was more active during the retrieval than during the acquisition phase. In the revised manuscript, we clarified that all ACA paths involved in the acquisition or expression of contextual fear responses were differentially activated between acquisition and expression of fear memory and are unlikely to provide a condition to produce a state-dependent effect on contextual fear memory (see Discussion, pg. 26, lines 13 – 16).

2) The presence of the experimenter during the acquisition session was a concern shared by the reviewers. It was unclear if the experimenter was present only during acquisition or also during test. If the former, was the position consistent? If the experimenter was not present during test then the authors need to rule out that the presence/absence of the experimenter from acquisition to test did not influence the performance of the mice. This would require additional experiments.

In the revised manuscript, we clarified that the experimenter was present in the experimental room during the habituation phase, predator exposure, and context exposure. In all conditions, the experimenter’s position in the experimental room remained consistent (see Methods, pg. 31, lines 1 – 3).

3) Having the laser on for 5 min is long and some evidence that there were no tissue damaging effects would be of value. A control for this stimulation at some timepoint between acquisition and test in the control group is also needed.

In the new version of the manuscript, we provide a histological analysis of the tissue at the optic fiber’s tip after the behavioral procedure. As shown in the Nissl-stained sections, no tissue damage is seen in the areas surrounding the optic fiber’s tips, as evidenced by the lack of gliosis or neuronal loss (see Appendix – Figure 11). Note that the controls for optic stimulation comprised the experimental groups injected with AAV5-hSyn-mCherry—the control groups not expressing halorhodopsin (HR; controls)—that were stimulated continuously for 5 min with a yellow laser (589 nm) during either the acquisition or the expression phases.

4) The efficiency of terminal inhibition using eNpHR3.0 is relatively low. Please use slice recording to confirm that the terminal inhibition can reduce spiking evoked EPSCs of postsynaptic cells and that it does not lead to rebound excitation upon laser off.

In the new version of the manuscript, we provide a histological analysis of the tissue at the optic fiber’s tip after the behavioral procedure. As shown in the Nissl-stained sections, no tissue damage is seen in the areas surrounding the optic fiber’s tips, as evidenced by the lack of gliosis or neuronal loss (see Appendix – Figure 11). Note that the controls for optic stimulation comprised the experimental groups injected with AAV5-hSyn-mCherry—the control groups not expressing halorhodopsin (HR; controls)—that were stimulated continuously for 5 min with a yellow laser (589 nm) during either the acquisition or the expression phases.

5) Additional behavioural measures such as freezing during the test and escape-like responses should be coded and presented.

As presented in the Appendix – Table 1 and discussed in the main text (see Discussion, pg. 22, lines 13 – 17), mice exhibited no freezing responses during exposure to the predatory context. Moreover, the revised manuscript clarifies that our protocol conditions yielded only occasional escape responses. Thus, only a few escape episodes were noted during cat exposure, which occurred mostly when the mice became aware of the cat and fled back to Box 1; furthermore, only occasional escape episodes were seen by the mice during exposure to the predatory context (see Appendix – Behavioral Protocol). Considering that the mice did not display any contextual freezing and only engaged in a minimal number of escape attempts, these responses were not coded.

6) Do the same ACA cells project to BLA, PERI, POST and PAG? It is likely that these projections originate from ACA cells at different cortical layers. The FG tracing experiment should provide some clues. Please use dual retrograde tracing to address the question.

Considering that the effects on acquisition or expression of contextual defensive responses were found in the ACA projections to the basolateral amygdala (BLA), perirhinal area (PERI), and dorsolateral periaqueductal gray (PAGdl), we performed triple retrograde tracing in the same animal to investigate the distribution of ACA neurons projecting to the BLA, PERI, and PAGdl. In four animals, three different retrograde tracers were ipsilaterally injected, namely, the retrograde AAV-Retro-Ef1a-mcherry- IRES-Cre, Fluoro Gold, and subunit B of the cholera toxin (CTb) in the PAGdl, BLA, and PERI, respectively (see Appendix – Triple retrograde labeling).

There is a layering pattern of the ACA neurons projecting to BLA, PERI, and PAGdl. Our analysis revealed that the ACA neurons projecting to the PAGdl are located predominantly in infragranular layers (layers V and VI; Appendix Figure 9E, G), and neurons projecting to the PERI are located predominantly in the supragranular layers (layers II and III; Appendix Figure 9F, G). The ACA neurons projecting to the BLA were found mainly in supragranular layers (layers II and III), but retrogradely labeled cells in the layer V were also occasionally found (Appendix Figure 9D, G). No double-labelling was found in the ACA > PAGdl projecting neurons (Appendix Figure 9G). Conversely, 51.66% and 46.98% of the ACA cells projecting to the BLA and those projecting to the PERI, respectively, were double-labeled (Appendix Figure 9G–I). These data suggest that nearly 50% of the ACA cells provided a branched pathway to the BLA and PERI.

These results are presented in the Appendix – Triple retrograde labeling and commented on the Discussion (pg. 25, lines 1 – 4, and pg. 26, line 9 – 12).

7) For the double FG-Fos double labeled cells, please perform further statistical analysis by comparing the actual overlap with chance level overlap based on Fos% and FG%. This helps understand whether there is a preferential overlap between Fos and FG.

In the revised version of the manuscript, the observed FG/Fos immunofluorescence double-labelling was compared to chance-level overlap by calculating a double-to-chance ratio as follows: *double labelling* = (number of FG/Fos double-labeled cells)/(total number of DAPI-positive cells); *chance level* = (number of FG-labeled cells/total number of DAPI-positive cells) × (number of Fos-labeled cells/total number of DAPI-positive cells); *double-to-chance ratio* = *double labelling/chance level* (see Methods – pg. 37, lines 5 – 16). These results were incorporated into previous chi-square tests to compare the proportion of FG/Fos double-labeled cells between the cat exposure condition and the predatory context condition (see Results – pg. 13, lines 20 – 23; pg. 16, lines 1 -5; pg. 18, lines 1 – 6; pg. 20, lines 8 – 12; and Methods – pg. 37, lines 5 – 16).

Interpretation.1) The lack of effect in the ACA>POST pathway inhibition study is taken as evidence that this pathway is not involved. Was the manipulation of this pathway insufficient to obtain an effect; or is it due to the relatively weak projection? The FG+Fos data speak to this somewhat, but those data raise the question with regard to the location of the optical fibers. When there is greater overlap the position of the optical fibers may be less of a concern as it is more likely to target some of those overlapping cells, but this likelihood is decreased in the lower percentage overlap. The histological data may shed light on this issue. Please provide a quantification of the terminal fields in BLA, PERI, POST and PAG.

To address this question, we first examined the position of the optic fibers, which was consistent across the experiments. In the revised manuscript, we illustrate the fiber optic position for 5 animals from the HR+ group that received optogenetic silencing of the ACA > POST pathway during cat exposure (see Appendix Figure 10). Next, we quantified the ACA terminal fields in the BLA, PERI, PAGdl, and POST. We selected three cases that received AAV5-hSyn-eNpHR3.0-mCherry injection into the ACA, and we used the software Fiji (Image J) to measure the density of terminals in the ACA projection fields. The corrected total field fluorescence (CTFF) was estimated and revealed the densest ACA projection field in the BLA, followed by the projections to the PERI, PAGdl, and POST, which contained the weakest projections from the ACA in all cases examined (see in Appendix, Quantification of ACA terminal fields and Figure 8). Note that the functional tracing also revealed that the ACA > POST pathway is weakly activated during both the acquisition and expression phase of contextual fear memory. Therefore, these observations (i.e., consistency of optic fiber position, relatively weak projection from the ACA to the POST, and low activation of this pathway during both phases of contextual fear memory) support the findings that silencing the ACA > POST projections had no effect on contextual fear memory of a predatory threat.

2) Additional controls are required for the Fos study. These include: Fos expression when the animal is exposed to a non-threat but novel stimulus; Fos expression when the animal is exposed to a novel context or a context paired with a non-threat stimulus. This will help uncover whether the ACA response is specific to threat and threat paired context.

In the ACA, DAPI and Fos quantification were performed in four conditions: exposure to the cat, the predatory context, the novel non-threat stimulus (plush cat), and the context paired with non-threat stimulus. Accordingly, approximately 44% and 28% of the neurons during exposure to the cat and cat-related context, respectively, expressed Fos protein. Conversely, Fos-labeled cells in response to exposure to the plush cat and plush cat context were approximately17% and 11%, respectively. Statistical analysis revealed that ACA Fos expression in response to the cat and cat-related context was significantly higher relative to its expression level in response to exposure to a novel non-threat stimulus and to the context paired with a non-threat stimulus (see in the Appendix, Comparison of ACA activity and Figure 3; and Discussion – pg. 22, lines 18 – 24).

Impact. The lack of female mice is problematic from rigor, translation, and impact perspectives. At minimum please provide a discussion regarding the potential effect of sex on these studies.

To the best of our knowledge, all reports examining the neural pathways underlying anti-predator responses to natural threats have been carried out in male rodents (rats and mice), and the present study is a follow-up investigation to reveal core circuits implicated in contextual fear. Conceivably, and as discussed in the revised version of the manuscript, the basic circuits organizing these responses should be similar in both genders. However, gender differences are expected in terms of the responsivity of elements of these circuits, especially considering that critical sites in these circuits, such as the hypothalamus, hippocampus, and amygdala, are responsive to gonadal steroid hormones (Simerly et al., 1990). In line with this view, studies have shown higher baseline levels of anti-predator responses in female rodents (Blanchard et al. 1989, 1992). Therefore, further studies are needed to investigate gender differences in the responsivity of circuits mediating contextual fear memory to predatory threats (see Discussion – pg. 26, lines 32 – 33 and pg. 27, lines 1 – 7).

[Editors' note: further revisions were suggested prior to acceptance, as described below.]

The reviewers acknowledge that the revised manuscript and the added data constitute a considerable improvement. However, three concerns raised previously were judged to be inadequately addressed. These are outlined below.1) The response to the state dependent hypothesis is unclear. To clarify the concern, state dependency occurs when information needs to be in the same state at retrieval as during acquisition in order to be recalled. Examples of what may constitute a state include drug influence, emotional states, the activation or inhibition of specific brain area, or pathway etc. This needs to be addressed in the manuscript.

State dependency occurs when information needs to be in the same state at retrieval as during acquisition in order to be recalled. Considering the pathways involved in acquisition or expression of contextual fear in the present investigation, we argued that all pathways involved in memory acquisition or expression were differentially activated during each stage and they are not expected to generate a state-dependent condition that affects memory. Thus, the pathways that affect memory encoding (i.e., the AM > ACA, ACA > BLA and ACA > PERI pathways) were more active in the encoding rather than retrieval phase. Conversely, the pathway that influenced memory expression (i.e., the ACA > PAG pathway), was more active during the retrieval than during the acquisition phase. However, regardless of endogenous activity patterns, there is a hypothetical possibility of a disruptive effect if memory is encoded in the absence of a given pathway and tested in its presence. In this case, it is important to bear in mind that as much as the optogenetic silencing of a given pathway is effective, it is unlikely to provide a complete inactivation of the entire pathway (see Discussion, pg. 26, lines 15 – 22).

2) The authors offer a descriptive response of the other behaviours observed in their task. The reviewers ask that the authors provide a quantification of those behaviours in the manuscript despite their low frequency.

For freezing responses, we showed and discussed that mice exhibited no freezing responses during exposure to the predatory context (see Discussion, pg. 22, lines 13 – 17, and Appendix Table 1), and we do not see the use of showing this lack of freezing in both control and experimental groups in the graphs. For escape-like responses, as we discussed (see Appendix – Behavioral Protocol), our protocol conditions yielded only occasional escape responses. During cat exposure, most of the animals flee away from the cat as they notice its presence and present no more clear escape responses during the predator exposure test (PET). During exposure to the predatory context, the animals presented only occasional escape responses, and several animals from the control groups do not even show clear escape responses remaining most of the time risk assessing the environment (see Appendix – Behavioral Protocol). Therefore, in our experimental conditions, escape-like response is a poor index to probe defensive responses, and this is the reason why we did not include in the analysis. Nevertheless, for the sake of transparency, in the Appendix Table 2, we provide the escape responses for all behavioral experiments.

3) The response to lack of sex-dependent examination was considered insufficient. Since all studies are following up on prior work, there is a logical flaw in using this argument to exclude females because it would prevent females from ever being included in investigations following from work conducted in males. Moreover, the topic investigated here is of high relevance to both males and females and thus inclusion of females is important for the rigor and impact of the work. The authors are asked to provide behavioural data from the task using females. These data must be carefully contrasted to those observed in males and similarities and differences in behaviour and behavioural strategies must be explored and discussed. This is the bare minimum expected. A neurobiological analysis of the effect in females, of course, would strengthen the paper in orders of magnitude. In addition, the term "sex" should be used in place of "gender" because gender is not a construct that can be examined in non-human animals.

In the revised version, we provided a comparison between male and female mice on the behavioral responses in the predator exposure test (PET) and Context. We observed that male and female mice present similar defensive responses during exposure to the predator or predatory context (Appendix Table 1). Conceivably, the basic circuits organizing these responses should be similar in both sexes. In line with this view, a recent study carried out in male and female mice examining the role of the dorsal premammillary nucleus – a key element of the hypothalamic predator-responsive circuit – on coordination of anti-predatory responses did not report differences between sexes (Wang et al., 2021). However, sex differences are expected in terms of the responsivity of elements of defensive circuits, especially considering that critical sites in these circuits, such as the hypothalamus, hippocampus, and amygdala, are responsive to gonadal steroid hormones (Simerly et al., 1990). Therefore, we pointed out that further studies are needed to investigate sex differences in the responsivity of circuits mediating contextual fear memory to predatory threats (see Discussion, pg. 27, lines 6 – 17).

4) The response to the ACA-POST projection question regarding whether the pathway is involved or whether it is merely a weak projection does not seem to be addressed in a way that clarifies the concern.

For the lack of effect in the ACA > POST pathway inhibition, first we considered that the position of the optic fibers was consistent across the experiments and suitable to silence the pathway (see Results, pg. 17, lines 43 – 45). In favor of the lack of effect of the ACA > POST pathway, we showed that this pathway is weakly activated during both the acquisition and expression phase of contextual fear memory. We believe that these are the main points that give support to our findings that the ACA > POST is not involved in either the acquisition or expression of contextual fear memory. Finally, we also revealed that the ACA > POST pathway was the weakest compared to the other paths shown to influence fear memory (see Discussion, pg. 25, lines 7 – 17).

[Editors' note: further revisions were suggested prior to acceptance, as described below.]

The manuscript has been improved but there are some remaining issues that need to be addressed, as outlined below:1) The response on state-dependency did not have page numbers and lines that corresponded to appropriate text in the paper or the response in the letter did not align with the wording in the text. This needs to be clarified.

Please note that there is a misalignment in the converted Article pdf file from the uploaded Article docx file. For the response on the state – dependency we noted in the manuscript that “all ACA paths involved in the acquisition or expression of contextual fear responses were differentially activated between acquisition and expression of fear memory and are unlikely to provide a condition to produce a state dependent effect on contextual fear memory. However, regardless of endogenous activity patterns, there is a hypothetical possibility of a disruptive effect if memory is encoded in the absence of a given pathway and tested in its presence. In this case, it is important to bear in mind that as much as the optogenetic silencing of a given pathway is effective, it is unlikely to provide a complete inactivation of the entire pathway.” (see Discussion, pg. 26, lines 15 – 22).

2) Thank you for adding the behavioural data for the females in the supplemental. No neural data were added, which leaves the authors speculating about possible lack of differences at the circuit level based on lack of differences in behaviour. The authors need to acknowledge that a similar behavioural effect does not necessarily mean that the underlying circuits are the same. A more careful discussion of this should be added along with a statement about the need for future investigation into this matter and a more clear acknowledgement of the limitations of the exclusion of females from the neural investigations of this study.

In the revised version of the manuscript we clarified that regardless of similar innate and contextual fear responses seen in males and females, the underlying circuits are not necessarily the same. We have also discussed that in the literate, there is a lack of investigation on the neural circuits mediating anti-predatory responses in female rodents, and this is certainly a limitation of the present investigation. We further considered that a recent study carried out in male and female mice examining the role of the dorsal premammillary nucleus – a key element of the hypothalamic predator-responsive circuit – on coordination of anti-predatory responses did not report differences between sexes (Wang et al., 2021). However, sex differences are expected in terms of the responsivity of elements of defensive circuits, especially considering that critical sites in these circuits, such as the hypothalamus, hippocampus, and amygdala, are responsive to gonadal steroid hormones (Simerly et al., 1990). At this point, further studies are needed to investigate sex differences in the circuits mediating contextual fear memory to predatory threats. (see Discussion, pg. 27, lines 7 – 19).